# Interaction Measures, Partition Lattices and Kernel Tests for High-Order Interactions

**Zhaolu Liu**[1]    **Robert L. Peach**[2,3]    **Pedro A.M. Mediano**[4]    **Mauricio Barahona**[1]*

[1]Department of Mathematics, Imperial College London, United Kingdom
[2]Department of Neurology, University Hospital Würzburg, Germany
[3]Department of Brain Sciences, Imperial College London, United Kingdom
[4]Department of Computing, Imperial College London, United Kingdom

## Abstract

Models that rely solely on pairwise relationships often fail to capture the complete statistical structure of the complex multivariate data found in diverse domains, such as socio-economic, ecological, or biomedical systems. Non-trivial dependencies between groups of more than two variables can play a significant role in the analysis and modelling of such systems, yet extracting such high-order interactions from data remains challenging. Here, we introduce a hierarchy of $d$-order interaction measures, increasingly inclusive of possible factorisations of the joint probability distribution, and define non-parametric, kernel-based tests to establish systematically the statistical significance of $d$-order interactions. We also establish mathematical links with lattice theory, which elucidate the derivation of the interaction measures and their composite permutation tests; clarify the connection of simplicial complexes with kernel matrix centring; and provide a means to enhance computational efficiency. We illustrate our results numerically with validations on synthetic data, and through an application to neuroimaging data.

## 1   Introduction

There is increasing evidence that pairwise relationships are insufficient to model many real world systems [1, 2, 3]. The relevance of high-order interactions has been emphasised in many contexts, as relationships within social [4], ecological [5, 6], and biological systems [7, 8] frequently involve groups of three or more agents, beyond pairwise associations. Such high-order interactions can be neither trivially represented by a linear combination of dyadic relationships, as the presence of high-order interactions can significantly impact the dynamics on networked systems [9, 10, 11, 12, 13, 14, 15], nor can they be simply detected by joint independence tests which inherently ignore other possible factorisations of the joint distribution.

Direct measurements of group interactions are seldom available, leading to their omission, partial representation as projected pairwise interactions [16, 17, 18], or limited recovery through inference methods [19, 20]. Often, only indirect measurements of underlying interactions in real-world complex systems are available, in the form of $iid$ or time-series data. Methods that learn high-order interactions from time-series data have been developed [11, 21], however, their reliance on heuristic measures makes their interpretation difficult. Alternative methods have been developed with strong foundations in statistics [22] and information theory[23, 24, 25], however, as we show later, these are often based

---

*Corresponding author: m.barahona@imperial.ac.uk

37th Conference on Neural Information Processing Systems (NeurIPS 2023).

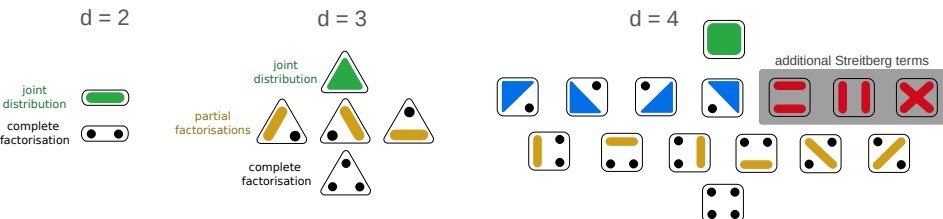

Figure 1: **Factorisations of joint distributions** $\mathbb{P}_{1\ldots d}$ **for** $d = 2, 3, 4$**.** The black dots indicate the marginal distributions of the singletons. The line, triangular and square shapes represent the joint distribution of two, three and four variables respectively. Different factorisations are presented as partitions of the $d$ variables ordered by increasing cardinality from top to bottom, so that all the factorisations with the same number of independent blocks appear at the same level. Joint independence considers only the top and bottom levels for each $d$, whilst the Lancaster interaction considers all terms except those in the shaded region. The Streitberg interaction considers all partitions. Hence, for $d = 2$, we have $\Delta_I^2\mathbb{P} = \Delta_L^2\mathbb{P} = \Delta_S^2\mathbb{P}$; whereas for $d = 3$, we have $\Delta_I^3\mathbb{P} \neq \Delta_L^3\mathbb{P} = \Delta_S^3\mathbb{P}$, and for $d \geq 4$, we have $\Delta_I^d\mathbb{P} \neq \Delta_L^d\mathbb{P} \neq \Delta_S^d\mathbb{P}$.

on an incomplete set of interactions and thus fail to capture all possible factorisations of the joint probability distribution [26, 27].

Kernel-based hypothesis tests provide a non-parametric, statistically robust framework for detecting relationships between variables from observational data. Such tests have been implemented for pairwise [28, 29, 30, 31] and $d$-variate joint independence [32, 33], and proven to be effective for non-trivial dependencies such as in the 3-way Lancaster test [34, 35]. However, to the best of our knowledge, cases where the number of variables exceeds 3 are still unexplored.

Here, we extend the capability for detection of high-order interactions by introducing a family of tests based on factorisations of the joint probability distribution that generalise systematically to any order $d$. At the head of this family, we introduce the Streitberg interaction test, which captures all factorisations of the joint distribution of order $d$. We further show that, despite the fact that the naive extension to $d \geq 4$ of the Lancaster interaction excludes some of the factorisations of the joint distribution, not all is lost, and we detail the conclusions that can be drawn from rejecting the $d$-order Lancaster interaction. Furthermore, we show that: (*i*) lattice theory provides a coherent theoretical foundation for detecting high-order interactions; (*ii*) interaction measures can be systemically derived from partition lattices; (*iii*) the corresponding Hilbert-Schmidt norm for kernel embeddings can naturally be extracted from the product lattice; (*iv*) the composite permutation tests can be performed with regard to the second level of the lattice; and (*v*) the lattice formulation allows us to propose a generalised interaction measure that can be used to test whether a given factorisation can be factorised further. Despite the inescapable combinatorial nature of testing high-order interactions, we offer approaches to reduce the computational complexity of our $d$-order interaction tests informed by our links with lattice theory. Finally, we present numerical validations of our tests on synthetic data before applying them to real-world neuroimaging data.

## 2 Interaction Measures

The most basic form of interaction between variables occurs between two variables, $X^1$ and $X^2$, and is often characterized by the lack of pairwise independence, i.e., the difference between the joint distribution $\mathbb{P}_{X^1,X^2}$ and the product of the marginal distributions $\mathbb{P}_{X^1}$ and $\mathbb{P}_{X^2}$ (see Fig. 1, $d = 2$). Let us consider $d$ random variables $\{X^1, X^2, \ldots, X^d\}$ with joint distribution $\mathbb{P}_{X^1,\ldots,X^d} =: \mathbb{P}_{1\ldots d}$ (for readability, hereafter we use this subscript notation). The $d$ variables are jointly independent if and only if $\mathbb{P}_{1\ldots d}$ can be factorised as the product of the univariate marginal distributions. Hence we can introduce an interaction measure that vanishes only when the variables are jointly independent:

$$\Delta_I^d\mathbb{P} = \mathbb{P}_{1\ldots d} - \prod_{i=1}^{d} \mathbb{P}_i. \tag{1}$$

When $d > 2$, however, the criterion for joint independence fails to capture high-order interactions, since additional factorisations must be considered. For instance, if $\mathbb{P}_{123}$ could be factorised as

$\mathbb{P}_1\mathbb{P}_{23}$ (see Fig. 1, $d = 3$), joint independence would be rejected, yet there is no 3-way interaction. Instead, we seek a way of identifying high-order interactions for $d$ variables where all lower order independencies can be rejected.

One approach to address the shortcomings of joint independence in identifying high-order interactions for $d > 2$ variables is to use a signed measure called the Lancaster interaction [34]. The Lancaster interaction for $d = 3$ is defined as $\Delta_L^3\mathbb{P} = \mathbb{P}_{123} - \mathbb{P}_1\mathbb{P}_{23} - \mathbb{P}_2\mathbb{P}_{13} - \mathbb{P}_3\mathbb{P}_{12} + 2\mathbb{P}_1\mathbb{P}_2\mathbb{P}_3$. $\Delta_L^3\mathbb{P}$ vanishes if $\mathbb{P}_{123}$ can be factorised in any way, and has been implemented as a kernel-based test statistic to identify non-factorisable joint distributions [34, 35]. Lancaster also generalised this measure to the multivariate case with $d$ variables:

$$\Delta_L^d\mathbb{P} = \prod_{i=1}^{d} \left(\mathbb{P}_i^* - \mathbb{P}_i\right), \tag{2}$$

where $\mathbb{P}_i^*\mathbb{P}_j^* \cdots \mathbb{P}_k^* = \mathbb{P}_{ij\cdots k}$. The Lancaster interaction (2) vanishes when the joint distribution can be factorised into jointly independent subvectors for $d = 3$, but it fails when $d \geq 4$ [26]. Specifically, $\Delta_L^4 P$ does not vanish if $\mathbb{P}_{1234}$ factorises into $\mathbb{P}_{12}\mathbb{P}_{34}$, $\mathbb{P}_{13}\mathbb{P}_{24}$, or $\mathbb{P}_{14}\mathbb{P}_{23}$. Despite this, the Lancaster interaction is not entirely uninformative for $d \geq 4$, and in Section 4.1 we examine the necessary conditions for it to vanish.

The desired vanishing property can be achieved by using the Streitberg interaction [26], a more general interaction measure defined using partitions. Let $D$ be the set of random variables $\{X^1, X^2, \ldots, X^d\}$, and let $\Pi(D)$ denote the set of all partitions of $D$, where a partition $\pi$ is a collection of nonempty, pairwise disjoint subsets (blocks) $b_j \subseteq D$ that cover $D$. Then, the Streitberg interaction measure is defined as

$$\Delta_S^d\mathbb{P} = \sum_{\pi \in \Pi(D)} (|\pi| - 1)!\,(-1)^{|\pi|-1}\mathbb{P}_\pi. \tag{3}$$

Here, $|\pi|$ denotes the cardinality of the partition $\pi$, and $\mathbb{P}_\pi = \prod_{j=1}^{r} \mathbb{P}_{b_j}$ is the corresponding factorisation with respect to $\pi = b_1|b_2|\ldots|b_r$. It has been proven that $\Delta_S^d\mathbb{P} = 0$ if the joint distribution can be factorised in any way, although the converse is not true in general [34].

# 3 Partition Lattices

The expressions of the interaction measures outlined in the previous section can be systematically generated from partition lattices. Interestingly, this formulation also allows us to establish further theoretical links with simplicial complexes.

**Notation.** A partially ordered set defined on a set $S$ with a binary relation $\leq$ is a lattice $\mathcal{L}$ if for any $\sigma, \pi \in \mathcal{L}$ there exists a greatest lower bound (meet) $\sigma \wedge \pi$ and least upper bound (join) $\sigma \vee \pi$ [36]. We denote the maximum and minimum element of a lattice as $\hat{1}$ and $\hat{0}$. Let $\Delta(\cdot)$ denote a real-valued function defined on $S$, then the sum function $f(\cdot)$ is analogous to the integration of $\Delta(\cdot)$ over the interval $[\hat{0}, \pi]$, where $\pi \in \mathcal{L}$. Then $\Delta(\cdot)$ can be expressed as the inverse operation of $f(\cdot)$:

$$f(\pi) = \sum \zeta(\sigma, \pi)\,\Delta(\sigma), \qquad \Delta(\pi) = \sum_{\sigma \leq \pi} \mu(\sigma, \pi)\,f(\sigma), \tag{4}$$

where the partial order is encoded by the Zeta matrix, with $\zeta(\sigma, \pi) = 1$ if $\sigma \leq \pi$ and 0 otherwise. Its inverse is the Möbius matrix with elements $\mu(\sigma, \pi)$ (for details see Section D), which can be obtained explicitly [37].

**Lattices and Interaction Measures.** The construction of the interaction measures in Section 2 is closely related to the partition lattice [26, 37]. The partition lattice is defined on the set $\Pi(D)$ with ordering given by the notion of partition refinement. A partition $\sigma$ is said to refine another partition $\pi$, denoted as $\sigma \preceq \pi$, if every block of $\sigma$ is fully contained within a block of $\pi$. The lattice structure thus allows us to define the least upper bound $\sigma \vee \pi$ and the greatest lower bound $\sigma \wedge \pi$ between any two partitions $\sigma$ and $\pi$. Clearly, the maximum element of the partition lattice, $\hat{1}$, corresponds to the joint distribution, whereas the minimum element, $\hat{0}$, corresponds to the complete factorisation (Fig. 1).

Within this formalism, the interaction measures are obtained when the sum function $f(.)$ in (4) is the probability distribution function. It then follows that the interaction measures are obtained

from the inverse operation. Indeed, the Streitberg interaction (3) is given by the Möbius inversion defined over the complete partition lattice. In contrast, the Lancaster interaction is obtained when the inversion is defined over the subset of partitions that have at most one non-singleton block, which we denote as the *Lancaster sublattice*. In other words, the sum function associated with the Lancaster interaction considers fewer interactions than the Streitberg interaction and its Möbius inversion has correspondingly fewer terms, as seen in (2). Note that for $d = 3$, the full (Streitberg) lattice and the Lancaster sublattice are the same, hence the interaction measures coincide. For $d \geq 4$, however, the Streitberg and Lancaster lattices, and consequently the corresponding interaction measures, are different, as shown by the extra partitions with two non-singleton blocks (shaded region) in Figure 1. Note also that joint independence considers a trivial sublattice with only two elements: $\hat{0}$ and $\hat{1}$ (for any $d$). The Möbius inversion on this sublattice leads to (1), which only vanishes for complete factorisations.

**Links to Simplicial Complexes.** A popular representation of high-order systems in the literature is through simplicial complexes [38]. Importantly, the simplicial complex construction can also be understood in terms of partition lattices. In particular, the elements in a $(d-1)$-simplex have inclusion ordering and thus form a subset lattice, e.g., $\{X^1, X^2\}$ is a subset of $\{X^1, X^2, X^3\}$. The subset lattice has been utilised in the understanding of information geometry [39] and solving tensor balancing [40]. It can be shown that the deatomised sublattice (with removal of singletons) is isomorphic to the Lancaster lattice [41]. Furthermore, the boundary matrices of a $(d-1)$-simplex appear as block matrices both in the Zeta matrix and the (inverse) Möbius matrix of the $d$-order partition lattice. These are explored further in Section E in the SI.

Note that the non-singleton partitions, i.e., those that do not belong to the Lancaster sublattice, are not in the set of simplicial complexes. Hence these blocks and their respective refinements *cannot* be expressed in terms of boundary matrices. Therefore the full partition lattice and its resulting Streitberg interaction provides more information compared with measures originating from sublattices, and in particular the Lancaster lattice and its related simplicial complex construction [23, 24]. Whilst simplicial complexes, and similarly the Lancaster interaction, can be recursively constructed from lower order elements, this is not possible for the Streitberg interaction due to the partitions without singletons, which has implications on computational efficiency (Section 5).

# 4 Kernel Interaction Tests

The interaction measures in Section 2 can be utilised as statistics in non-parametric tests when embedded into reproducing kernel Hilbert spaces (RKHS). Given a symmetric, positive definite function $k^i : \mathcal{X}^i \times \mathcal{X}^i \to \mathbb{R}$, there is an associative RKHS $\mathcal{H}^i$ with the reproducing kernel property. For $X^i \in \mathcal{X}^i$, we denote $\phi^i(\cdot)$ as the canonical map of $k^i(X^i, \cdot)$. The kernel mean embedding of $\mathbb{P}_{X^i}, \mu_{\mathbb{P}_{X^i}}$, satisfies $\mathbb{E}_{X^i} f(X^i) = \left\langle f, \mu_{\mathbb{P}_{X^i}} \right\rangle_{\mathrm{HS}}$, where HS stands for Hilbert-Schmidt. If the kernel is characteristic, the mean embedding is injective and the norm of the signed measure is zero if and only if the measure is zero itself [28, 34, 42]. These properties enable us to create meaningful non-parametric tests by computing the kernel mean embedding of desired interaction measures.

In this section, we extend interaction measures for random variables to the $d$-variate case, and formulate a family of interaction tests including the Streitberg interaction, the Lancaster interaction, joint independence and a generalised interaction.

For proofs of the propositions, please see Appendix A.

## 4.1 Lancaster Interaction

Let us first consider the Lancaster interaction. Although it does not necessarily vanish when $d \geq 4$ for all types of factorisations due to the lack of certain partitions, here we find the necessary conditions for the Lancaster interaction to vanish.

**Proposition 1.** *If the joint distribution $\mathbb{P}_{1\ldots d}$ can be factorised into $\mathbb{P}_{\pi_v}$ where $\pi_v$ are partitions with at least one singleton, then $\Delta_L^d \mathbb{P} = 0$.*

*Remark:* Note that $\mathbb{P}_{\pi_v}$ is a broader set of partitions compared to the set of partitions in the Lancaster lattice, e.g., for $d = 5$, $\mathbb{P}_{12}\mathbb{P}_{34}\mathbb{P}_5$ satisfies Proposition 1, yet it is not a constituent of the Lancaster lattice, which only consists of partitions with at most one non-singleton.

We now define the Mixed Central Moment Operator as the kernel embedding of the Lancaster interaction, which follows immediately from the expansion of (2) [27]:

**Definition 1** (Mixed Central Moment Operator).

$$\mathcal{M}_d = (-1)^{n-1}(n-1)\,\mathbb{E}\left[\prod_{i=1}^{d}\phi^i\right] + \sum_{\pi_\ell \neq \hat{0}}(-1)^{|\pi_\ell|-1}\,\mathbb{E}\left[\prod_{s}\phi^s\right]\prod_{j}\mathbb{E}\left[\phi^j\right], \qquad (5)$$

*where $\pi_\ell$ are the set of partitions with at most one non-singleton (i.e., those that belong to the Lancaster lattice), $s$ are the singletons, and $j$ runs over variables in the non-singleton block.*

**Proposition 2.** *By rearranging, the Mixed Central Moment Operator can be simplified to:*

$$\mathcal{M}_d = \mathbb{E}\left\{\prod_{i=1}^{d}\left[\phi^i - \mathbb{E}[\phi^i]\right]\right\}. \qquad (6)$$

*Remark:* This simplification transparently re-expresses the operator as a central moment, instead of the complex sum of moments in (5). The simplification eliminates all partitions except $\hat{1}$.

We now define the entries in matrix $K_{ab}^i = k^i(x_a^i, x_b^i)$ for $iid$ samples $x_a^i$ and $x_b^i$ where $1 \leq a, b \leq n$ and $\tilde{K}^i = HK^iH$ where $H = I - \frac{1}{n}\mathbf{1}\mathbf{1}^\top$ is the centring matrix. The norm of the embedding above can serve as a test statistic. The estimator of the test statistic is derived as the V-statistic:

**Proposition 3** (Lancaster interaction estimator.).

$$||\hat{\mathcal{M}}_d||_{\mathcal{HS}}^2 = \frac{1}{n^2}\sum_{a=1}^{n}\sum_{b=1}^{n}\prod_{i=1}^{d}\tilde{K}_{ab}^i, \qquad (7)$$

## 4.2   Streitberg Interaction

Similarly we define the Mixed Cumulant operator in terms of kernel embeddings from the Streitberg interaction as:

**Definition 2** (Mixed Cumulant operator).

$$\mathcal{K}_d = \sum_{\pi \in \Pi(D)}(|\pi|-1)!(-1)^{|\pi|-1}\prod_{b \in \pi}\mathbb{E}\left\{\prod_{i \in b}\phi^i\right\}, \qquad (8)$$

*where $i$ is an element in block $b$ of partition $\pi$ in partition lattice $\Pi(D)$.*

This operator can be simplified in a similar way:

**Proposition 4.**

$$\mathcal{K}_d = \sum_{\pi_s \in \Pi(D)}(|\pi_s|-1)!(-1)^{|\pi_s|-1}\prod_{b \in \pi_s}\mathbb{E}\left\{\prod_{i \in b}\left[\phi^i - \mathbb{E}[\phi^i]\right]\right\}, \qquad (9)$$

*where $\pi_s$ are the partitions with no singletons in partition lattice $\Pi(D)$.*

*Remark:* Note that partitions with singletons are eliminated after centring, and the remaining partitions without singletons form an upper semi-lattice. The relationship between the Mixed Cumulant operator and the Mixed Central Moment operator is given by:

**Lemma 1.** *The Mixed Cumulant operator is equal to the sum of Mixed Central Moment Operator products associated with partitions with no singletons denoted as $\pi_s$ [27]:*

$$\mathcal{K}_d = \sum_{\pi_s \in \Pi(D)}(|\pi_s|-1)!(-1)^{|\pi_s|-1}\prod_{b \in \pi_s}\mathcal{M}_b. \qquad (10)$$

*Remark:* When $d = 2, 3$, $\mathcal{K}_d$ and $\mathcal{M}_d$ are identical, which further confirms the moment-cumulant relationship.

*Remark:* From the partition lattice, $\{\hat{0}, \hat{1}\} \subseteq \pi_\ell \subseteq \pi$, where $\pi_\ell$ is the set of partitions in the Lancaster lattice, and $\pi$ is the full set in the Streitberg interaction. When $d = 2$, we have $\{\hat{0}, \hat{1}\} = \pi_\ell = \pi$, and when $d = 3$, we have $\{\hat{0}, \hat{1}\} \subset \pi_\ell = \pi$.

The estimation using a V-statistic for the norm of the Streitberg interaction is given by:

**Proposition 5** (Streitberg interaction estimator).

$$||\hat{\mathcal{K}}_d||^2_{\mathcal{HS}} = \sum_{\pi_s,\pi'_s \in \Pi(D)} (|\pi_s|-1)!(|\pi'_s|-1)!(-1)^{|\pi_s|+|\pi'_s|} \frac{1}{n^{|\pi_s|+|\pi'_s|}} \sum_{\{b_i\}}\sum_{\{b'_i\}}\prod_{i=1}^{d}\tilde{K}^i_{b_i,b'_i}, \quad (11)$$

*where $1 \leq b_i, b'_i \leq n$ and $|\{b_i\}| = |\pi_s|$, $|\{b'_i\}| = |\pi'_s|$. i.e. indices $b_i = b_j$ if $i,j$ are in the same block in $\pi_s$ similarly for $b'_i$. The sets of indices $\{b_i\}, \{b'_i\}$ are disjoint. An example for $d = 4$ is given in the SI.*

**Norms and the Product Lattice.**    The test statistics of the interaction measures, i.e., the norm of the interaction operators, can also be directly obtained from the Möbius inversion of the Cartesian product of the upper semilattice, thus avoiding the need to compute the square of the operator. The Cartesian product of two lattices with product ordering is also a lattice [43]. The elements in the Cartesian product are analogous to the inner products of kernel embeddings in the estimators. Therefore, the coefficients can be computed directly from the product lattice, as shown in Appendix B.

### 4.3   Description of the Statistical Tests

The Streitberg interaction test involves rejecting the null hypothesis that the joint distribution can be factorised in some way, which occurs when $\Delta^d_S \mathbb{P} \neq 0$. Similarly, for the Lancaster interaction test, we reject the null hypothesis that the joint distribution can be factorised into partitions with at least one singleton by checking $\Delta^d_L \mathbb{P} \neq 0$. Both null hypotheses consist of multiple sub-hypotheses, and an example for $d = 3$ is discussed in Ref. [34, 35]. Importantly, the number of sub-hypotheses to be tested is related to factorisations with only two blocks, i.e., those that form the second level from the top in the partition lattice (see Figure 1). In the case of rejecting some but not all the sub-hypotheses, please see Appendix C for the discussion.

To test each sub-hypothesis $\mathbb{P}_{1...d} = \mathbb{P}_b\mathbb{P}_{b'}$, for efficiency we fix the observations of the variables in the largest block, and permute the observations of the remaining variables in the other block to induce the independence structure in the null distribution. We then use the Monte-Carlo approximation to compute the p-value [32], and apply a simple correction[35] generalised to $d$ variables to correct for multiple hypothesis testing. Finally, we reject the composite null hypothesis if all corresponding sub-hypotheses are rejected. To achieve this, we employ the permutation test outlined in Algorithm 1.

---

**Algorithm 1** Permutation test for the interaction measures

---

test-statistic $\leftarrow ||\hat{\mathcal{K}}^d||^2_{(\tilde{K}^1,...,\tilde{K}^d)}$
**for** $i \in \{\pi\}$ (here $\pi = b|b'$) **do**
   initialise empty $P$-dimensional vector **T**
   **for** $p = 1:P$ **do**
      **for** $j = 1:d$ **do**
         **if** $j \in \underset{x \in \{b,b'\}}{\operatorname{argmin}}(|x|)$ **then**
            $\tilde{K}^j \leftarrow \tilde{K}^j_{(s)}$ {kernel matrices after random permutations on the observations of $X^j$}
      $T[p] \leftarrow ||\hat{\mathcal{K}}^d||^2_{(\tilde{K}^1,...,\tilde{K}^d)}$
   tmp $\leftarrow \#\{p \in \{1,...,P\}|\mathbf{T}[p] \geq$ test-statistic$\}$
   pval $\leftarrow (\text{tmp}+1)/(P+1)$ {Monte-Carlo approximation [32]}
   **if** pval $> \alpha$ **then**
      reject $\leftarrow 0$ {'simple correction' [35]}
      **break**
reject $\leftarrow 1$
**return** reject

---

By rejecting the null hypothesis of the Streitberg interaction test, it follows that we reject the null hypothesis of the Lancaster interaction test, as well as the null hypothesis of joint independence. The set of sub-hypotheses in joint independence is a subset of the set in the Lancaster interaction test, which is in turn a subset of the set in the Streitberg interaction test. The number of sub-hypotheses for an interaction measure is equal to the number of elements on the second level of the respective lattice.

For Streitberg interaction, this number is $(2^{d-1} - 1)$ as follows from the full partition lattice; for Lancaster interaction, this number is $d$ due to the reduced Lancaster lattice; for joint independence the number of sub-hypotheses stays fixed as the corresponding 2-element lattice always contains precisely one element, i.e., $\hat{0}$ corresponding to $\mathbb{P}_1 \cdots \mathbb{P}_d$.

*Remark:* Although our focus is on composite interaction tests to unveil high-order interactions by leveraging the vanishing of Streitberg and/or Lancaster interactions, these measures are also valuable to test other lower-order independence hypotheses wherein factorisations can also lead to vanishing interaction measures. For example, both Lancaster and Streitberg statistics can be implemented to test for joint independence (as shown in Section 6), and for marginal independence.

### 4.4 Generalised Interaction Measure

The measures described so far are only able to handle interactions within one group of variables. But what if we are interested in knowing whether a factorisation (e.g., $\mathbb{P}_{12}\mathbb{P}_{34}$) factorises further? To answer this, we can simply formulate an interaction measure based on a partition [44]. Because an interval of a lattice $\mathcal{L}$ is a subset of the form $[\sigma, \pi] = \{x \mid \sigma \leq x \leq \pi\}$ and is also a lattice [43], the interval lattice can be used to produce generalised interactions for multiple groups of variables:

**Definition 3** (General interaction operator)**.**

$$\Delta_S^{\pi_s}\mathbb{P} = \sum_{\sigma \leq \pi_s} m(\sigma, \pi_s)\mathbb{P}_\sigma = \prod_{b \in \pi_s} \Delta_S^b\mathbb{P}, \tag{12}$$

*where $\pi_s$ are the partitions with no singletons, $m$ are the Möbius coefficients, and $b$ are the blocks in the partition $\pi_s$.*

Here, we only consider blocks of size at least 2, since a factorisation cannot not be defined for singletons. It is then clear that $\Delta_S^{\pi_s}\mathbb{P} = 0$ if any $\Delta_S^b\mathbb{P}$ is zero, and the rejection statement on each group will be dependent on their own vanishing conditions. This provides a general framework to detect interactions within blocks of variables. The corresponding kernel-based tests can be constructed by estimating the norm of $\Delta_S^{\pi_s}\mathbb{P}$. In fact, the generalised Streitberg interaction measure can be expressed in terms of ordinary Streitberg interaction measures defined in Section 4.2 [44]:

$$\Delta_S^{\pi_s}\mathbb{P} = \sum_{\sigma \vee \pi_s = \hat{1}} \prod_{b \in \sigma} \Delta_S^b\mathbb{P}. \tag{13}$$

## 5  Computational Considerations

The time complexity of computing the Lancaster interaction estimator is $\mathcal{O}(dn^2)$, where $d$ is the order of interaction and $n$ is the number of samples. This follows immediately after centring the kernel matrices, since $\hat{1}$ is the only partition with no singletons. In contrast, the Streitberg interaction has a larger number of partitions, given by the Bell number, $B_d$ [45]. However, by simplifying the computation through centring, we can reduce this to the number of partitions with no singletons, given by $F_d$ [46]. While this number is still combinatorial, it is significantly smaller compared to $B_d$ (see Table 1 in Appendix G).

The resulting estimator of the Streitberg interaction criterion then consists of $F_d$ inner products. The summation indices of the inner products come from both partitions and are at most $d$ (when $d$ is even) or $d-1$ (when $d$ is odd). Whilst the summations can be contracted, the choice of contraction ordering plays a crucial role in the time complexity [47, 48, 49]. By computing the summations related to one partition in parallel, since the index sets from $\pi_s$ and $\pi'_s$ are disjoint, the overall complexity can be reduced to $\mathcal{O}(n^{min(|\pi_s|,|\pi'_s|)+1})$ (see Table 2 in Appendix 5). For example, the inner product between the kernel embeddings for $\mathbb{P}_{1234}$ and $\mathbb{P}_{12}\mathbb{P}_{34}$ is $\frac{1}{n^3}\sum_i\sum_j\sum_k K_{ij}^1 K_{ij}^2 K_{ik}^3 K_{ik}^4$, and an improved contraction ordering is $\frac{1}{n^3}\sum_i[\sum_j K_{ij}^1 K_{ij}^2][\sum_k K_{ik}^3 K_{ik}^4]$.

The time complexity of the Streitberg interaction estimator can be further reduced in a recursive setting. Specifically, if $\pi_s \vee \pi'_s \prec \hat{1}$, then the associative inner product can be expressed as a product of independent sums that can be found in lower order interactions. For example, $\langle \mu_{\mathbb{P}_{12}\mathbb{P}_{34}}, \mu_{\mathbb{P}_{12}\mathbb{P}_{34}} \rangle$ can be decomposed as $\langle \mu_{\mathbb{P}_{12}}, \mu_{\mathbb{P}_{12}} \rangle \langle \mu_{\mathbb{P}_{34}}, \mu_{\mathbb{P}_{34}} \rangle$ which would have already been computed for the pairwise interactions of $\{X^1, X^2\}$ and $\{X^3, X^4\}$. This means that if one has already computed

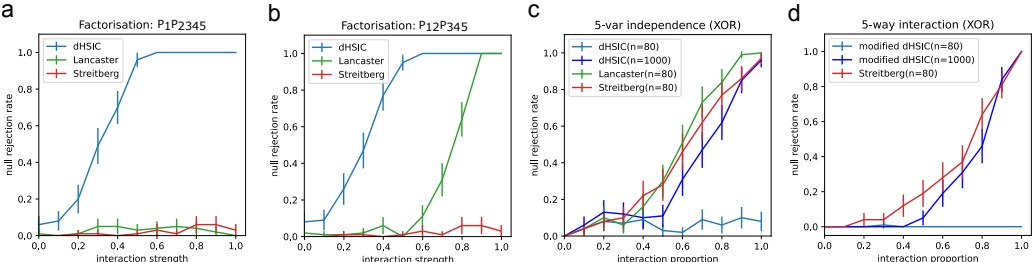

Figure 2: **High-order tests on synthetic data.** (a) dHSIC is unable to detect the marginal $\mathbb{P}_1\mathbb{P}_{2345}$ factorisation. (b) Only the Streitberg test is able to capture the factorisation without singletons $\mathbb{P}_{12}\mathbb{P}_{345}$. (c) Testing joint independence of a 5-way XOR example, Lancaster and Streitberg tests achieve higher power than dHSIC for the same number of samples ($n$=80). Only by increasing the number of samples to $n$=1000 does dHSIC display comparable power. (d) The modified dHSIC test requires substantially more samples than the Streitberg test to display comparable power when testing all sub-hypotheses of the interaction in the XOR example.

the inner products of $d - 2$ variables (incremental by 2 since the smallest block is at least size 2), then the only remaining inner products to compute are between those embeddings associated with partitions $\pi_s$ and $\pi'_s$ such that $\pi_s \vee \pi'_s = \hat{1}$. In other words, the computation of inner products for high-order interactions can be reduced to the computation of lower order interactions, as long as the corresponding partitions can be joined to form a partition of all $d$ variables. For further discussions on computational complexity, see Appendix G.

## 6 Experiments

We first validate our family of $d$-order interaction tests (in particular, $d = 5$) on synthetic datasets with ground truth interactions, and then investigate their application to a neuroimaging dataset. Unless stated otherwise, the significance level is set to $\alpha = 0.05$, sample size to $n = 80$, and we use Gaussian kernels with the median heuristic as the bandwidth.

**Synthetic Experiments: Multivariate Gaussians.** We first compare results from applying the dHSIC test (developed solely to detect joint independence) [32], and the Lancaster and Streitberg interaction tests to five-variable multivariate Gaussian distributions $\mathcal{N}(\mu, \Sigma)$ with mean $\mu$ and covariance $\Sigma$. For the first example, $\mu = [0, 0, 0, 0, 0]$ and $\Sigma_1 = [[1, 0, 0, 0, 0], [0, 1, \beta, \beta, \beta], [0, \beta, 1, \beta, \beta], [0, \beta, \beta, 1, \beta], [0, \beta, \beta, \beta, 1]]$ where $0 \leq \beta \leq 1$ defines the interaction strength between variables and consequently $\mathbb{P}_{12345} = \mathbb{P}_1\mathbb{P}_{2345}$. For the second example, we have the same $\mu$ and $\Sigma_2 = [[1, \beta, 0, 0, 0], [\beta, 1, 0, 0, 0], [0, 0, 1, \beta, \beta], [0, 0, \beta, 1, \beta], [0, 0, \beta, \beta, 1]]$ and hence $\mathbb{P}_{12345} = \mathbb{P}_{12}\mathbb{P}_{345}$.

Figure 2(a-b) shows that dHSIC fails to capture the partial factorisation of the joint distribution in both cases. Whilst the Lancaster interaction test is able to detect the factorisation in the first experiment (Figure 2(a)), it fails on the second experiment where the factorisation contains no singletons (Figure 2(b)). The Streitberg interaction test is the only one that detects both factorisations.

**Synthetic Experiments: 5-Way XOR Gate.** Next, we investigate the power of the tests using a data set constructed using an XOR gate. We generate $n$ samples of $V, W, X, Y, Z \overset{i.i.d.}{\sim} \mathcal{U}(0, 4)$, $Z_{:i} = (V_{:i} + W_{:i} + X_{:i} + Y_{:i}) \mod 4$ and $Z_{i+1:n} \sim \mathcal{U}(0, 4)$) (where the samples $[i + 1 : n]$ act as noise). We then gradually increase the interaction proportion, $0 \leq i/n \leq 1$. By construction, this dataset does not contain pairwise, 3-way or 4-way interactions, and the 5-way interaction becomes increasingly easier to detect as $i/n$ approaches 1.

Given that the joint distribution in this example cannot be factorised, all tests should reject joint independence. We show in Figure 2(c), that the rejection rates of joint independence for Streitberg and Lancaster approach one as the interaction level increases for a small number of samples ($n = 80$), whereas dHSIC displays low power for the same number of samples. Only by increasing the number

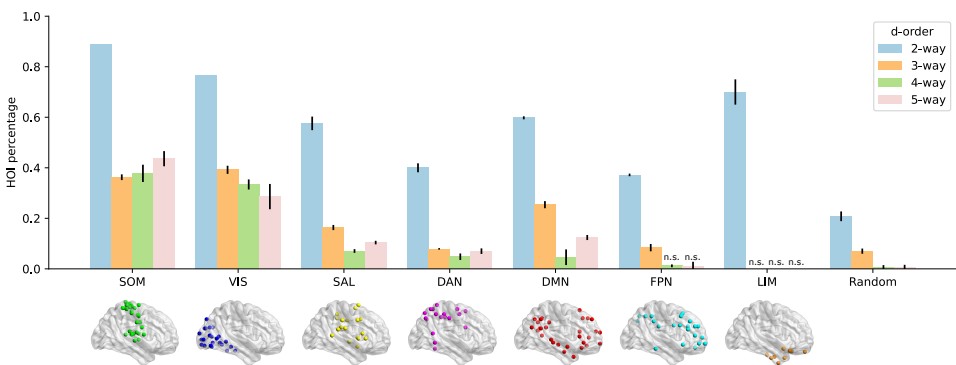

Figure 3: **Concentration of high-order interactions in brain resting state networks.** The percentage of rejected null hypotheses for $d$-order ($d = 2, 3, 4, 5$) interaction tests, when the $d$ regions are selected within each of seven brain resting state networks (RSNs) or drawn at random across the brain. For each order $d$, we carry out Fisher-exact tests for over-representation for each RSN *vs.* random. All Fisher tests are significant ($p$-value$< 0.05$) except where shown as (n.s.). Abbreviations for RSNs: DMN: default mode network, SOM: somatomotor, VIS: visual, SAL: salience, DAN: dorsal attention network, FPN: frontoparietal network, LIM: limbic.

of samples to $n = 1000$ do we observe comparable performance for dHSIC. To test for 5-way interactions in the XOR data set, we omit the Lancaster test which cannot detect factorisations into non-singletons. We thus compare the Streitberg interaction test with a modified dHSIC [34, 35] that tests each individual sub-hypothesis (e.g., when testing $(X, Y) \perp\!\!\!\perp (Z, W)$ it is sufficient to treat $(X, Y)$ and $(Z, W)$ as single variables and perform joint independence tests for two variables). We find again that the modified dHSIC requires a much larger sample size to achieve comparable performance to that of the Streitberg interaction test (Fig. 2(d)).

To investigate how the test power degrades when the sample size is decreased, we performed further experiments in Fig 8 in Appendix F. In all cases, we find that the performance of dHSIC degrades faster than that of Lancaster and Streitberg, i.e., the dHSIC null rejection rate decays substantially faster than that of Lancaster and Streitberg as the sample size decreases.

**Neuroimaging Dataset.** As a proof of concept of applications to real data, we apply the Streitberg interaction test to detect high-order interactions in brain activity data. The dataset consists of resting-state fMRI data from 50 unrelated subjects part of the Human Connectome Project [50, 51]. For each subject, the data consists of 100 time series capturing the activity of each of the regions in the Schaefer brain atlas [52]. Importantly, these 100 regions can be divided in 7 groups called 'Resting State Networks' (RSN) [53]. Experimental evidence has shown that regions within the same RSN perform similar functions [52], so we hypothesised that high-order interactions would be more probable within RSNs.

To avoid dealing with issues of non-stationarity in the time-series data, we first take temporal averages to obtain the mean activity per region and subject (see Appendix H for preprocessing steps). We then perform the Streitberg interaction test to sets of brain regions that are either taken from within the same RSN or sampled at random from the whole brain. In each case, we sample 500 sets of regions or take all possible combinations, whichever is lowest.

The results in Figure 3 align with our general hypothesis: 2-, 3-, 4- and 5-way interactions are significantly more common among regions belonging to the same RSN as compared to random sets of regions, confirming that our interaction test can successfully detect high-order interactions in real-world data. Notably, our results show a larger percentage of 2-way interactions in the SOM and VIS compared to regions such as the FPN or DAN, suggesting that there is increased redundancy in the system relative to these other regions. The SOM and VIS are structurally coupled, modular sensorimotor processing regions, that benefit from information redundancy to increase robustness. Moreover, the proportion of high-order interactions (3-, 4- and 5-way) to 2-way interactions is larger in SOM and VIS compared to FPN or DAN, potentially reflecting the differing balance of redundancy and synergy in these brain regions [54]. Additional analyses are shown in Appendix H, but the

neuroscience implications of these results, including observed variation across regions, will be the object of future work.

## 7   Discussion

In this work, we have established formal connections between lattice theory and a family of high-order interaction measures based on factorisations of the joint probability distribution. The link to lattice theory facilitates the derivation and interpretation of the measures; highlights the connection to simplicial complexes and other sublattices; and helps formulate the associated permutation tests more efficiently. We have shown empirically that the Lancaster test statistic is a good substitute for dHSIC to test for joint and marginal independence, whilst the Streitberg test statistic is able to better capture all factorisations of the joint distribution. Our work offers a rigorous and systematic approach for the reconstruction of high-order systems, but has some limitations. The computation of the Streitberg interaction estimator can be expensive and although we have proposed strategies to bring down the time complexity, the number of terms remains combinatorial (see Appendix G for discussions on practical limitations). In addition, our theoretical results rely on $iid$ data, a strong assumption in many real-world applications. This leaves several open directions. In particular, future work will investigate whether the null distribution of $\Delta_S^d P$ can be jointly approximated without the need for sub-hypotheses, thus reducing computational cost; how it can be accurately approximated when the data has temporal dependence; and what role high-order interactions may play in causal discovery.

## Acknowledgements

MB acknowledges support by the EPSRC under grant EP/N014529/1 funding the EPSRC Centre for Mathematics of Precision Healthcare at Imperial, and by the Nuffield Foundation under the project "The Future of Work and Well-being: The Pissarides Review". RP acknowledges funding from the Deutsche Forschungsgemeinschaft (DFG, German Research Foundation) Project-ID 424778381-TRR 295.

For the experiments in Section 6, data were provided by the Human Connectome Project, WU-Minn Consortium (Principal Investigators: David Van Essen and Kamil Ugurbil; 1U54MH091657) funded by the 16 NIH Institutes and Centres that support the NIH Blueprint for Neuroscience Research; and by the McDonnell centre for Systems Neuroscience at Washington University.

The authors would like to thank Jianxiong Sun, Xing Liu and Paul Expert for valuable discussions, Asem Alaa for help in maintaining the GitHub repository, and Andrea I. Luppi for assistance with the preprocessing of neuroimaging data.

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

# Appendix to "Interaction Measures, Partition Lattices and Kernel Tests for High-Order Interactions"

## A    Proofs

### A.1    Proof of Proposition 1

First we prove the following:

**Lemma 2.** *If* $\mathbb{P}_{1\cdots d} = \mathbb{P}_i \mathbb{P}_{1\cdots i-1, i+1 \cdots d}$ *for arbitrary* $i$, *then* $\Delta_L^d \mathbb{P} = 0$

*Proof.* Without loss of generality let $i = 1$. To simplify the notation, we adopt a shorthand convention for expression partitions. For example, a partition $\{\{X^1, X^3\}, \{X^2\}, \{X^4\}\}$ can be written as $13|2|4$. We denote the partition $1|23 \cdots d$ as $\pi$. Notice that the factorisation only consists of an extraction of a singleton, i.e. factorise one variable out of the joint distribution. The Möbius coefficients [41] for the partial partition lattice above are,

$$\mu(\sigma, \hat{1}) = \begin{cases} (-1)^{d-1}(d-1) & \text{if } \sigma = \hat{0} \\ (-1)^{|\sigma|-1} & \text{otherwise} \end{cases}.$$

The partial partition lattice is constructed as

$$123 \cdots d$$
$$1|23 \cdots d \qquad 2|13 \cdots d \qquad 3|12 \cdots d \qquad ...$$
$$1|2|3 \cdots d \qquad 1|3|2 \cdots d \qquad ... \qquad 2|3|1 \cdots d \qquad ...$$
$$\vdots$$
$$1|2|3| \cdots |d$$

Given $\mathbb{P}_{1\cdots d} = \mathbb{P}_1 \mathbb{P}_{2 \cdots d}$, the partial partition lattice associated with Lancaster interaction collapses into the following:

$$1|23 \cdots d$$
$$1|23 \cdots d \qquad 2|1|3 \cdots d \qquad 3|1|2 \cdots d \qquad ...$$
$$1|2|3 \cdots d \qquad 1|3|2 \cdots d \qquad ... \qquad 2|3|1 \cdots d \qquad ...$$
$$\vdots$$
$$1|2|3| \cdots |d$$

If a partition $\sigma$ is not a refinement of $\pi$, then it will be transformed to $\sigma \wedge \pi$, a partition one level down the lattice (as shown in the lattice above), e.g., let $\sigma$ be $2|13 \cdots d$, then $\sigma \wedge \pi$ is be $2|1|3 \cdots d$. When $\sigma$ is a refinement of $\pi$, then $\sigma \wedge \pi = \sigma$.

The transformed partitions will be cancelled out with the refinements of $\pi$ due to the fact that all partitions are counted exactly once and the partitions at two consecutive levels have alternating signs. On the $(d-1)$ level of the lattice, exactly $(n-1)$ partitions are not refinement of $\pi$ which are just enough to cancel out $(n-1)$ number of $\hat{0}$ at the bottom level.    □

When $\mathbb{P}_{1\cdots d} = \mathbb{P}_{\pi_s}$ where $\pi_s$ only contains non-singleton blocks, the Lancaster interaction

$$\Delta_L^d \mathbb{P} = \prod_{b \in \pi_s} \Delta_L^b \mathbb{P},$$

which is not necessarily zero unlike the Streitberg interaction. It will be zero, however, if any of the $\Delta_L^b \mathbb{P}$ is zero. If we factorise one singleton out of an arbitrary block $b$, $\pi_s$ will be a partition with one singleton block and hence $\Delta_L^b \mathbb{P} = 0$ by Lemma 2 above and so does $\Delta_L^d P$. Obviously if more singletons are factorised out, $\Delta_L^d P$ will remain zero.

## A.2 Proof of Proposition 2

Notice that this simplified Mixed Central Moment operator expression resembles the original Lancaster interaction in Equation (2) using the unique '∗ notation'. Notice that the resulting terms after expanding Equation (2) are distinctive with coefficient $(-1)^{|\pi_l|-1}$ depending on the except we have multiple occurrences of $\hat{0}$. This is due to the fact that $\mathbb{P}_i^*$ is the same as $\mathbb{P}_{\dot{i}}^*$. For example, when $d = 4$, instead of direct product of the marginals $\mathbb{P}_1\mathbb{P}_2\mathbb{P}_3\mathbb{P}_4$ that accounts for $\hat{0}$, there are also four other terms $\mathbb{P}_1^*\mathbb{P}_2\mathbb{P}_3\mathbb{P}_4$, $\mathbb{P}_1\mathbb{P}_2^*\mathbb{P}_3\mathbb{P}_4$, $\mathbb{P}_1\mathbb{P}_2\mathbb{P}_3^*\mathbb{P}_4$, $\mathbb{P}_1\mathbb{P}_2\mathbb{P}_3\mathbb{P}_4^*$ that are also equal to $\hat{0}$. Note that these terms have different signs comparing with $\mathbb{P}_1\mathbb{P}_2\mathbb{P}_3\mathbb{P}_4$ because of $\mathbb{P}_i^*$ and therefore result in $(-1)^{d-1}(d-1)$ in the Möbius function.

## A.3 Proof of Proposition 3

Computed immediately from the estimated norm of Mixed Central Moment operator, the Lancaster interaction estimator is

$$||\hat{\mathcal{M}}_d||_{HS}^2 = \left\langle \frac{1}{n} \sum_{a=1}^{n} \prod_{i=1}^{d} \tilde{\phi}^i(x_a^i), \frac{1}{n} \sum_{b=1}^{n} \prod_{i=1}^{d} \tilde{\phi}^i(x_b^i) \right\rangle$$

$$= \frac{1}{n^2} \sum_{a=1}^{n} \sum_{b=1}^{n} \prod_{i=1}^{d} \left\langle \tilde{\phi}^i(x_a^i), \tilde{\phi}^i(x_b^i) \right\rangle$$

$$= \frac{1}{n^2} \sum_{a=1}^{n} \sum_{b=1}^{n} \prod_{i=1}^{d} \tilde{K}_{ab}^i,$$

where $\tilde{\phi}(\cdot) = \phi(\cdot) - \mathbb{E}[\phi(\cdot)]$.

## A.4 Proof of Proposition 4

This can be completed by manual expansion and then equating the terms. Alternatively we can prove it more easily by the relationship of Lancaster interaction and Streitberg interaction in Lemma 1.

$$\mathcal{M}_b = \mathbb{E}\left\{ \prod_{i\in b} \left[\phi^i - \mathbb{E}[\phi^i]\right] \right\}$$

$$\mathcal{K}_d = \sum_{\pi_s \in \Pi(D)} (|\pi_s| - 1)!(-1)^{|\pi_s|-1} \prod_{b\in\pi_s} \mathcal{M}_b$$

$$= \sum_{\pi_s \in \Pi(D)} (|\pi_s| - 1)!(-1)^{|\pi_s|-1} \prod_{b\in\pi_s} \mathbb{E}\left\{ \prod_{i\in b} \left[\phi^i - \mathbb{E}[\phi^i]\right] \right\}.$$

## A.5 Proof of Proposition 5

Similar to the norm of the Mixed Central Moment operator, the norm of Mixed Cumulant operator can be obtained by simply squaring. Note that this can be computed directly using the product lattice discussed in Section 4.2.

$$||\hat{\mathcal{K}}_d||_{HS}^2$$

$$= \left\langle \sum_{\pi_s} (|\pi_s| - 1)!(-1)^{|\pi_s|-1} \frac{1}{n^{|\pi_s|}} \sum_{\{b_i\}} \prod_{i=1}^{d} \tilde{\phi}^i(x_{b_i}^i), \sum_{\pi_s'} (|\pi_s'| - 1)!(-1)^{|\pi_s'|-1} \frac{1}{n^{|\pi_s'|}} \sum_{\{b_i'\}} \prod_{i=1}^{d} \tilde{\phi}^i(x_{b_i'}^i) \right\rangle$$

$$= \sum_{\pi_s,\pi_s'} (|\pi_s| - 1)!(|\pi_s'| - 1)!(-1)^{(|\pi_s|-1)+(|\pi_s|-1)} \frac{1}{n^{|\pi_s|+|\pi_s'|}} \left\langle \sum_{\{b_i\}} \prod_{i=1}^{d} \tilde{\phi}^i(x_{b_i}^i), \sum_{\{b_i'\}} \prod_{i=1}^{d} \tilde{\phi}^i(x_{b_i'}^i) \right\rangle$$

$$= \sum_{\pi_s,\pi_s'} (|\pi_s| - 1)!(|\pi_s'| - 1)!(-1)^{|\pi_s|+|\pi_s|} \frac{1}{n^{|\pi_s|+|\pi_s'|}} \sum_{\{b_i\}} \sum_{\{b_i'\}} \prod_{i=1}^{d} \tilde{K}_{b_i,b_i'}^i$$

# B  Product Lattice

The partition lattice is reduced to an upper semilattice after centring. When $d = 4$, the reduced lattice only contains $\mathbb{P}_{1234}, \mathbb{P}_{12}\mathbb{P}_{34}, \mathbb{P}_{13}\mathbb{P}_{24}$ and $\mathbb{P}_{14}\mathbb{P}_{23}$ which are the 4 partitions with no singletons. Given the two arbitrary lattices $\mathcal{L}$ and $\mathcal{L}'$, the product lattice formulated using the Cartesian product of $S \times S'$ with product order can be defined. Given two elements $(\sigma, \sigma'), (\pi, \pi') \in \mathcal{L} \times \mathcal{L}'$, $(\sigma, \sigma') \le (\pi, \pi')$ if and only if $\sigma \le \pi$ and $\sigma' \le \pi'$. In our case specifically, $\mathcal{L}$ and $\mathcal{L}'$ are the identical reduced partition lattices of $d$ variables after centring, and the elements in the product lattice are analogous to the inner products when computing the kernel-based estimator. Below we show the product lattice at $d = 4$. Therefore, it can be utilised to directly compute the coefficients of the estimator in Proposition 5. Note that the order within each element is not particularly informative in this case as the kernel matrices are symmetric.

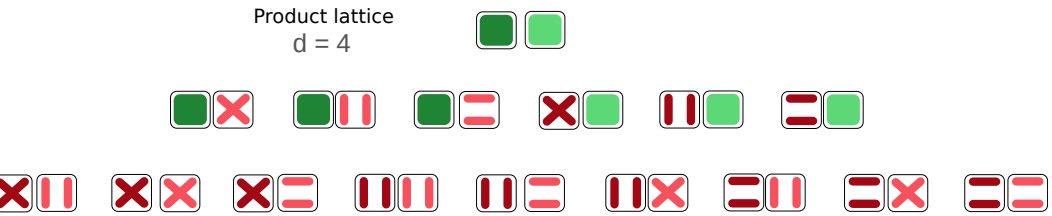

Figure 4: **Product lattice for $d = 4$.** The two lines and squares represent the non-singleton factorisations with two blocks and one block. Dark green and dark red highlight the elements in $\mathcal{L}$, whilst light green and light red are used for the elements in $\mathcal{L}'$.

The resulting estimator using centred kernels when $d = 4$ is

$$
\begin{aligned}
||\hat{\mathcal{K}}_4||_{\mathcal{HS}}^2 =& \frac{1}{n^2} \sum_{ij} \tilde{K}_{ij}^1 \tilde{K}_{ij}^2 \tilde{K}_{ij}^3 \tilde{K}_{ij}^4 \\
& - \frac{2}{n^3} \sum_{ijk} \tilde{K}_{ij}^1 \tilde{K}_{ij}^2 \tilde{K}_{ik}^3 \tilde{K}_{ik}^4 - \frac{2}{n^3} \sum_{ijk} \tilde{K}_{ij}^1 \tilde{K}_{ik}^2 \tilde{K}_{ij}^3 \tilde{K}_{ik}^4 - \frac{2}{n^3} \sum_{ijk} \tilde{K}_{ij}^1 \tilde{K}_{ik}^2 \tilde{K}_{ik}^3 \tilde{K}_{ij}^4 \\
& + \frac{1}{n^4} \sum_{ijkl} \tilde{K}_{ij}^1 \tilde{K}_{ij}^2 \tilde{K}_{kl}^3 \tilde{K}_{kl}^4 + \frac{1}{n^4} \sum_{ijkl} \tilde{K}_{ij}^1 \tilde{K}_{kl}^2 \tilde{K}_{ij}^3 \tilde{K}_{kl}^4 + \frac{1}{n^4} \sum_{ijkl} \tilde{K}_{ij}^1 \tilde{K}_{kl}^2 \tilde{K}_{kl}^3 \tilde{K}_{ij}^4 \\
& - \frac{2}{n^4} \sum_{ijkl} \tilde{K}_{ij}^1 \tilde{K}_{il}^2 \tilde{K}_{kj}^3 \tilde{K}_{kl}^4 - \frac{2}{n^4} \sum_{ijkl} \tilde{K}_{ij}^1 \tilde{K}_{il}^2 \tilde{K}_{kl}^3 \tilde{K}_{kj}^4 - \frac{2}{n^4} \sum_{ijkl} \tilde{K}_{ij}^1 \tilde{K}_{kl}^2 \tilde{K}_{il}^3 \tilde{K}_{kj}^4
\end{aligned}
$$

Notice that a few terms are summed up twice. This is due to the symmetry in the product lattice we described above.

# C  Composite Null Hypothesis

The permutation test strategy we propose for Streitberg interaction and Lancaster interaction is carried out via multiple sub-hypotheses, and the rejection of the interaction is confirmed once all the sub-hypotheses are rejected. Instead of having to consider all possible factorisations $(B_d)$ for the sub-hypotheses, by considering the lattice structure we only need to perform $(2^{d-1} - 1)$ sub-tests of factorisations corresponding to the 2nd level of the partition lattice, since all other sub-tests are consequences of these.

In the case of rejecting the Streitberg interaction, it won't necessarily be the case that all sub-hypotheses are rejected, allowing us to derive a more detailed understanding of how the joint distribution is factorised. For example, in the 7 sub-hypotheses for the Streitberg interaction when $d = 4$, there is a clear difference between the number of rejections when the ground truth factorisations are $\mathbb{P}_1\mathbb{P}_{234}$ or $\mathbb{P}_1\mathbb{P}_2\mathbb{P}_{34}$. The former results in 6 rejections while the latter results in only 4 rejections because $\mathbb{P}_1\mathbb{P}_2\mathbb{P}_{34}$ is the mutual refinement of $\mathbb{P}_{12}\mathbb{P}_{34}$, $\mathbb{P}_1\mathbb{P}_{234}$ and $\mathbb{P}_2\mathbb{P}_{134}$. Hence, even though the vanishing conditions of Streitberg interaction are not 'if and only if', we can still narrow down the range of the possible ground truth factorisations using the rejected sub-hypotheses.

The operations can be easily explained using the lattice. Before performing any sub-test, $\Pi(D)$ is the space of all possible ground truth factorisation. Whenever we reject a sub-hypothesis, the refinements of the associated partition $\pi$ gets eliminated. For example, if we reject $\mathbb{P}_1\mathbb{P}_{234}$, then we also automatically reject $\mathbb{P}_1\mathbb{P}_2\mathbb{P}_{34}$, $\mathbb{P}_1\mathbb{P}_3\mathbb{P}_{24}$, $\mathbb{P}_1\mathbb{P}_4\mathbb{P}_{23}$ and $\mathbb{P}_1\mathbb{P}_2\mathbb{P}_3\mathbb{P}_4$. In other words, partitions in the interval lattice $[\hat{0}, \pi]$ are eliminated. The remaining partitions until there are no further rejections are the only possible choices and also form a lattice. Again we illustrate this idea using the partition lattice of 4 variables (see Figure 5).

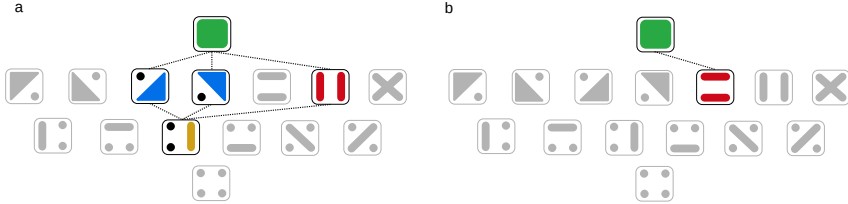

Figure 5: **Remaining factorisations with partial rejection of sub-hypotheses for** $d = 4$**.** The greyed out partitions represent the eliminated factorisations due to the rejections in the sub-tests. A dashed line between two partitions indicates the refinement ordering (only shown for those not rejected). (a) In the case that 4 of the sub-hypotheses are rejected, all of their refinements are automatically eliminated. The remaining elements form an interval lattice and is isomorphic to the partition lattice of 3 variables. (b) If 6 of the sub-hypotheses can be rejected, the ground truth can only be either the factorisation remaining or non-factorisable. Similarly this is isomorphic to the partition lattice of two variables.

_Remark:_ The simplification of the operators (computation of the test statistics) can be reflected in a collapsed partition lattice without singletons. However, one should not confuse this with the full lattice that is associated with the test procedures, i.e. one should always consider the full partition lattice to figure out the possible configurations of the ground truth factorisation based on what we outlined above. The two lattices have the same form however they have completely different implications. The lattice corresponding to test procedures can only be reduced as a result of the rejections. For example, 7 sub-tests are needed instead of just 3 tests that involve the factorisations without singletons when $d = 4$.

# D   Möbius Inversion

Instead of deriving the Möbius function as the inverse of the Zeta function, it can also be computed using the expression below [37],

$$\mu(\sigma, \pi) = \begin{cases} 1 & \text{if} \quad \sigma = \pi \\ -\sum_{\sigma \leq \rho < \pi} \mu(\sigma, \sigma) & \text{if} \quad \sigma < \pi \\ 0 & \text{otherwise} \end{cases}.$$

We can check the validity of this formula by computing the Möbius on the trivial partition lattice associated with joint independence, which contains two elements, $\hat{0}$ and $\hat{1}$,

$$\mu(\hat{1}, \hat{1}) = 1$$
$$\mu(\hat{0}, \hat{1}) = -\mu(\hat{1}, \hat{1}) = -1,$$

hence

$$\Delta_I^d \mathbb{P} = \mu(\hat{1}, \hat{1})\mathbb{P}_{1\cdots d} + \mu(\hat{0}, \hat{1})\prod_{i=1}^{d} \mathbb{P}_i$$

$$= \mathbb{P}_{1\cdots d} - \prod_{i=1}^{d} \mathbb{P}_i.$$

# E    Links to Simplicial Complexes

A $k$-simplex $\theta$ is a set of $k+1$ vertices $\theta = [p_0, ..., p_k]$, i.e., a 1-simplex is a line, a 2-simplex is a triangle, 3-simplex is a tetrahedron, etc. A simplicial complex $\Theta$ is a collection of simplices that satisfy two conditions: (i) if $\theta \in \Theta$, then all the sub-simplices $v \subset \theta$ built from subsets of $\theta$ are also contained in $\Theta$; and (ii) the non-empty intersection of two simplices $\theta_1, \theta_2 \in \Theta$ is a sub-simplex of both $\theta_1$ and $\theta_2$. The first condition makes the inclusion ordering in the subset lattice natural for simplicial complexes.

Below we have the Zeta matrix in Figure 6 and Möbius matrix in Figure 7 to illustrate that the boundary operators (with no directionality) of $(d\text{-}1)$-simplex can be founded as block matrices within them. The subset lattice doesn't include the non-simplicial terms and we have shown numerically that these terms are essentially in the synthetic experiment when $\mathbb{P}_{1234} = \mathbb{P}_{12}\mathbb{P}_{34}$.

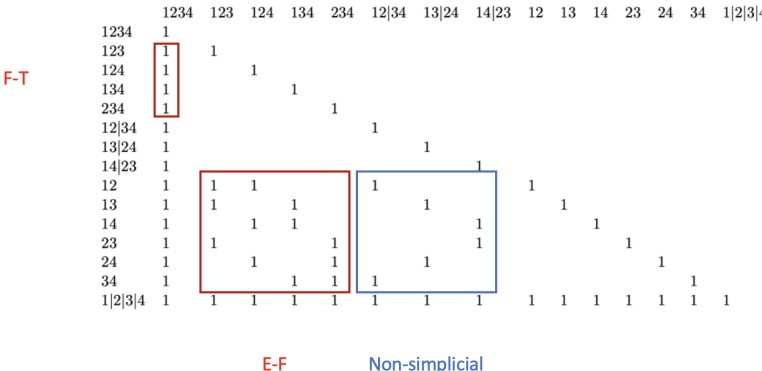

Figure 6: **Zeta matrix for the partition lattice of 4 variables.** Singletons in the partial factorisations are omitted for simplicity. The matrix can be decomposed into small blocks which correspond to the boundary operators in 3-simplex. Here the two red blocks represent the **E**dge-to-**F**ace operator and **F**ace-to-**T**etrahedron operator. The blue block corresponds to the non-simplicials, partitions with no singletons.

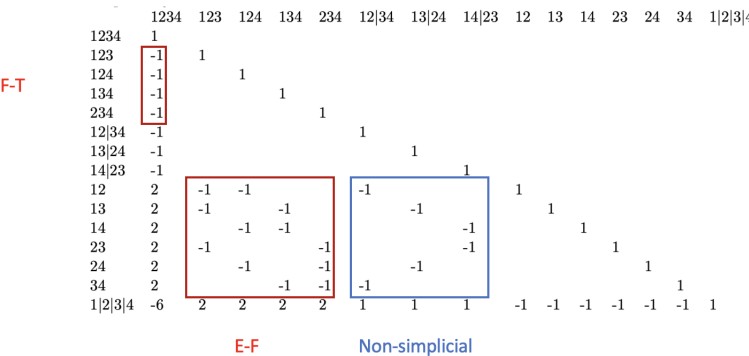

Figure 7: **Möbius matrix for the partition lattice of 4 variables.** Comparing with the Zeta matrix above we see that the block structure is preserved. Each block can be obtained using the Möbius inversion on the incidence matrix of its own partially ordered set, e.g. $\{12|34, 13|24, 14|23, 12|3|4, 13|2|4, 14|2|3, 23|1|4, 24|1|3, 34|1|2\}$ forms a partially ordered set. The values are always $-1$ since the partially order set only contains the elements from two neighbouring levels.

# F   Interplay of Test Power and Sample Size

To explore the effect of the different sample size on test power, we have performed four experiments, each with three interaction strengths, to show how a decrease in the sample size affects the null rejection rate (Fig 8). In all cases, the Lancaster and Streitberg tests are able to accurately detect higher order interactions with as low as 50-100 samples (depending on the interaction proportion/strength), whilst dHSIC requires substantially more samples. This provides further experimental evidence that the proposed Streitberg interaction test has superior sensitivity, as compared to dHSIC, in detecting higher order interactions.

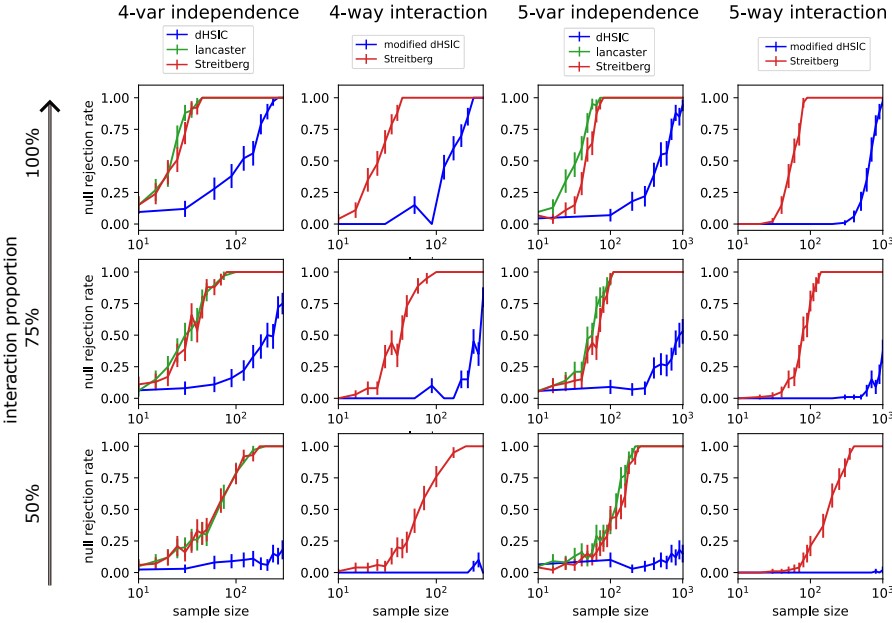

Figure 8: **Decay of accuracy rates as the sample size is decreased in the XOR example.** Independence tests (first and third columns); interaction tests (second and fourth columns). In all cases, dHSIC degrades substantially faster than Lancaster and Streitberg.

# G   Computational Complexity and Practical Limitations

## G.1   Computational Complexity

The original time complexity for computing the Streitberg interaction estimator is $\mathcal{O}(B_d^2 n^{2d})$ where $B_d$ is the Bell number, which represents the number of partitions in a set of cardinality $d$. Hence $B_d^2$ is the number of terms in the Streitberg interaction estimator. For fixed number of samples $n$, our lattice formulation allows us to reduce the number of terms in the mixed cumulant operator from $B_d$ to $F_d$. Table 1 illustrates this reduction in the number of terms before and after eliminating the partitions with singletons. After this reduction, the complexity becomes $\mathcal{O}(F_d^2 n^{2d})$.

Table 1: Number of terms in the test statistics before/after eliminating the partitions with singletons.

| No. of variables | 4 | 5 | 6 | 7 | 8 |
|---|---|---|---|---|---|
| Streitberg⇒(centred) | 15⇒(4) | 52⇒(11) | 203⇒(41) | 877⇒(162) | 4140⇒(715) |
| Lancaster⇒(centred) | 12⇒(1) | 27⇒(1) | 58⇒(1) | 121⇒(1) | 248⇒(1) |

For fixed $d$, we can further optimise the time complexity by optimal contraction ordering, using the fact that two index sets are disjoint, such that $\mathcal{O}(n^{2d})$ becomes $\mathcal{O}(n^{min(|\pi_s|,|\pi_s'|)+1})$. This reduction for different $d$ can be found in Table 2.

Table 2: Time complexity for $d$-order interaction estimators.

| $d$-order | dHSIC | Lancaster⇒(optimised) | Streitberg⇒(optimised) |
|-----------|-------|------------------------|-------------------------|
| 2-way | $\mathcal{O}(n^2)$ | $\mathcal{O}(n^2)$ | $\mathcal{O}(n^2)$ |
| 3-way | $\mathcal{O}(n^2)$ | $\mathcal{O}(n^2)$ | $\mathcal{O}(n^2)$ |
| 4-way | $\mathcal{O}(n^2)$ | $\mathcal{O}(n^8)\Rightarrow\mathcal{O}(n^2)$ | $\mathcal{O}(n^8)\Rightarrow\mathcal{O}(n^3)$ |
| 5-way | $\mathcal{O}(n^2)$ | $\mathcal{O}(n^{10})\Rightarrow\mathcal{O}(n^2)$ | $\mathcal{O}(n^{10})\Rightarrow\mathcal{O}(n^3)$ |

Similarly, the time complexity for computing the $d$-order Lancaster interaction estimator naïvely is $\mathcal{O}(2^{d+1}n^{2d})$. By centring, i.e. eliminating the partitions with singletons, the only element left is $\hat{1}$. Therefore the Lancaster interaction estimator can be computed in $\mathcal{O}(dn^2)$.

## G.2 Practical Considerations

The problem of detecting $d$-order interactions among a group of $e$ variables is combinatorial, as it entails checking groups of $d$ variables chosen from the $e$ variables. Clearly, such combinatorial problems become infeasible as $e$ and/or $d$ become large. Recent work on real data from various application areas has focused on revealing any interactions beyond pairwise, i.e., $d > 2$. It has been shown that interactions with $d = 3, 4$ already make a significant difference to the analysis of network structure and network dynamics [2, 21, 55], highlighting the potential benefits to be gained from tests that detect high order interactions.

Many well-recognised and widely used theoretical approaches only have closed forms for $d = 3$ [56, 57], and cannot be generalised to arbitrary $d$, since the number of terms in those approaches is related to the Dedekind number which becomes rapidly intractable [58]. In contrast, here we show that the Streitberg interaction can be explicitly defined for any $d$, and we have devised theoretical and computational strategies to reduce its computational cost via the lattice theory formulation.

In the case where high order interactions for a range of $d$ are of interest, it is possible to discount the computation of certain high order interactions when some lower-order interactions are present. If all the lower order interactions are absent, then testing the $d$-order interaction is the same as testing the joint independence for $d$ variables. For example when $d = 3$ and all pairwise interactions are absent, then testing the Streitberg interaction reduces to testing joint independence of 3 variables. More generally, if we test the interactions bottom-up, i.e., recursively from the lower orders upwards, the expression for Streitberg becomes simpler whenever there is a lower-order independence. The simplification is possible because the Streitberg interaction can be rewritten as the sum of the differences between a factorisation and the product of the marginals due to the fact that the sum of Möbius coefficients is zero. Hence the presence of a lower order independence allows us to simplify the $d$-order interaction formula.

Alternatively, there are also cost-reducing simplifications if we do the tests top-down (i.e. starting from order $d$ downwards), although the problem still remains combinatorial. If there are partial rejections for some tests involved in the $d$-order Streitberg test, we can narrow down the possible choices of the factorisation (as discussed in Appendix C). This allows us to eliminate the lower order factorisations that fall in the lattice branches of the rejected second level factorisations, thus reducing the total tests needed for the factorisation of the $d$-variables. Further simplifications are achieved by accounting for overlaps of the lattice branches.

All these observations can be taken into account in the construction of hypergraph representations, as further discussed below in Appendix H.

# H Neuroimaging Data

## H.1 Preprocessing

Preprocessing of neuroimaging data was performed following Luppi *et al.* [54]. We summarise the key steps here, and refer interested readers to the original paper for further details.

We started from the 'minimally preprocessed' release of the fMRI data from the Human Connectome Project [50, 51]. This data was preprocessed with bias field correction, functional realignment, motion

correction and spatial normalization to Montreal Neurological Institute (MNI-152) standard space with 2 mm of isotropic resampling resolution. We then removed the first ten points in the time series to avoid transient effects introduced by the scanner. Finally, we further denoised the data with the anatomical CompCor method, which involves regressing out potential noise confounds (specifically, five principal components of white matter activity, five principal components of cerebrospinal fluid activity, and 12 motion parameters including head translation, rotation, and their temporal derivatives). All preprocessing steps were performed using the CONN toolbox (`https://www.nitrc.org/projects/conn/`), version 17f58. The resulting volume was parcellated according to the Schaefer-100 atlas [52] by spatially averaging across all voxels in the same region for each timestep.

Table 3: Time taken for experiments in Figure 3.

|  | SOM | VIS | SAL | DAN | DMN | FPN | LIM | Random |
|---|---|---|---|---|---|---|---|---|
| 2-way | 1s | 2s | 1s | 2s | 2s | 2s | 1s | 12s |
| 3-way | 12s | 18s | 7 | 13s | 8s | 8s | 1s | 13s |
| 4-way | 5m36s | 5m | 2m44s | 2m31s | 2m | 2m10s | 1s | 2m41s |
| 5-way | 2h18m24s | 1h30m9s | 1h3m | 54m40s | 1h12m4s | 27m16s | 2s | 29m |

We report the computational time for all fMRI experiments in Table 3. All experiments carried out on a 2015 iMac with 4 GHz Quad-Core Intel Core i7 processor and 32 GB 1867 MHz DDR3 memory.

## H.2 Analysis of Simple Hypergraphs

As an illustration of future lines of work, we hierarchically constructed hypergraphs from the identified $d$-order interactions. Specifically, individual regions are nodes and the high-order interactions are used to define hyperedges, incrementally including the interactions of increasing order. The resulting hypergraphs integrate information across orders of interaction, and can be analysed by computing structural properties of hypergraphs. As an example, we show the degree assortativity of the different hypergraphs in Fig 9. We find that almost all RSNs display a negative degree assortativity (but much less so than random), and only FPN shows positive degree assortativity for higher order hypergraphs. Future analyses of these hypergraphs will study the links between high-order interactions in brain activity and different functional areas.

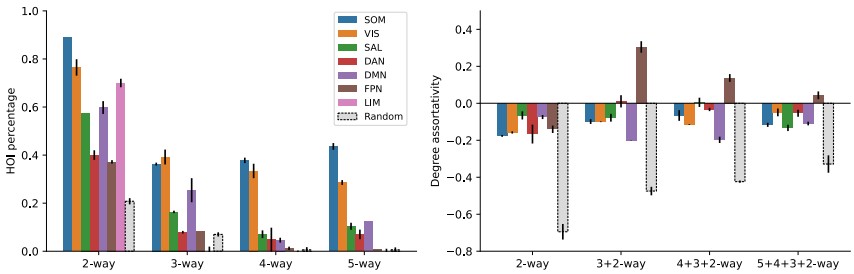

Figure 9: **Hypergraph analysis of the neuroimaging data.** Percentage of high-order interactions in RSNs (left) and degree assortativity of hypergraphs constructed from the high-order interactions detected (right).

## H.3 Alternative Hypergraphs: Emergent and Redundant High-Order Interactions

To interpret the presence of lower order interactions in larger cliques, one can alternatively build a hypergraph as follows: i) If a $d$-order hyperedge is detected and none of the lower order $(d-1)$-order hyperedges are present then we say that this $d$-order hyperedge reflects a purely synergistic (or emergent) interaction between the $d$ variables; ii) if the $d$-order and all the lower order interactions are present, then we say that the $d$-order interaction is purely redundant (and corresponds to a simplicial complex construction); iii) if some, but not all, lower order interactions are present (e.g., for 4 variables, only the 4-way interaction and one 3-way interaction are present), then both synergy and redundancy are present in this group of variables. Such a construction could offer an alternative,

statistically-motivated approach to computing synergy and redundancy, an important current topic of research in computational neuroscience, and its representation through hypergraphs.

# I Code

This code for performing the interaction tests and the synthetic experiments is provided in this anonymous Github repository `https://github.com/barahona-research-group/streitberg-interaction.git`.

