# OpenReview forum: "Interaction Measures, Partition Lattices and Kernel Tests for High-Order Interactions"
_NeurIPS.cc/2023/Conference — NeurIPS 2023 poster_

### Official Review · Reviewer_6qiu · 2023-07-08

**Soundness:** 3 good
**Presentation:** 3 good
**Contribution:** 3 good
**Rating:** 6
**Confidence:** 2

**Summary:**

The authors introduce a hierarchy of $d$-order interaction measures by introducing a family of tests based on factorizations of the joint probability distribution that generalize to any order $d$ and define non-parametric tests to establish the statistical significance of these d-order interactions. They link their approach to lattice theory which they use to reduce the computational complexity of our d-order interaction tests. They validate their method on a synthetic data set and through an application to a neuroimaging dataset.

**Strengths:**

The paper is very well written, covers an exciting topic and seems to introduce very original results that are of broad significance. The tests proposed and their relationship to lattice theory are fascinating and of some practical utility, in my opinion (although I must admit this work is outside my expertise).

**Weaknesses:**

As pointed out by the authors, for the time complexity of computing the Streitberg interaction estimator, the number of terms remains combinatorial. Yet clearly, as the authors show, it is possible to apply it to the HCP data. It would be useful to know the current practical limitations in terms of data size for the method proposed here.

**Questions:**

Can the authors comment more on the current practical limitations?
Can the authors better flesh out the consequences and interpretations of their results for the HCP data, as a means of demonstrating the utility and motivation for their approach?

**Limitations:**

The authors could explicitly add a "Limitations" subsection.

---

> ### Author Rebuttal · Authors · 2023-08-09
>
> We thank the reviewer for their comments and useful suggestions. We appreciate that one must consider both the overall complexity and the practical implementation of our tests. Therefore, we have provided a more in-depth analysis of both.
>
> First, in Table 2 in the main response we provide the overall time taken to compute the full experiments at each order for each brain network. Whilst the overall time to compute the tests increases with interaction order, it was still entirely feasible to run this on standard computer hardware, particularly given that 3-, 4-, and 5-way interactions will be sufficient for most datasets.
>
> Second, datasets often come with different sizes and so we further considered how sample size would affect our ability to identify high-order interactions. In the attached PDF, we Figure 1 shows how the rejection rate of our tests reduces as we decrease the sample size. The Lancaster and Streitberg tests are able to accurately detect higher order interactions with as low as 50-100 samples (depending on the interaction proportion/strength), whilst dHSIC requires substantially more samples.
>
> Finally, we report the big-O notation complexity in Table 1 in the main response for each interaction order. We show that our theoretical improvements, which follow from our mathematical links with lattice theory, reduce the time complexity significantly for $d>3$ interactions in both the Lancaster and Streitberg interaction estimators.
>
> Regarding the HCP data set, we believe there are some interesting interpretations and important implications. If we consider the SOM or VIS, our results show a larger percentage of 2-way interactions compared to regions such as the FPN or DAN, suggesting that there is increased redundancy in the system relative to these other regions. The SOM and VIS are structurally coupled, modular sensorimotor processing regions, that benefit from information redundancy to increase robustness. These findings are in line with recent work reporting higher redundancy in sensorimotor regions [1], although further investigation is needed to draw firm conclusions about brain function from these analyses.
>
> For further investigation, we hierarchically constructed hypergraphs from the identified interactions at different orders and then analysed their structure. By computing the degree assortativity of each hypergraph we found that most regions display a negative degree assortativity that became less negative with increasing order interactions. Additionally, we noticed that the degree assortativities when only considering pairwise connections d=2 are quite similar across regions. However, when including higher-order interactions in the hypergraphs, we find that the degree assortativities of each region diverge from each other. This has important implications on how the RSNs are defined if one considers high-order interactions.
>
> [1]  Luppi, A.I., Mediano, P.A.M., Rosas, F.E. et al. A synergistic core for human brain evolution and cognition. Nat Neurosci 25, 771–782 (2022).

---

### Official Review · Reviewer_vitW · 2023-07-09

**Soundness:** 4 excellent
**Presentation:** 4 excellent
**Contribution:** 4 excellent
**Rating:** 8
**Confidence:** 4

**Summary:**

The paper describes a kernel test for estimating d-th order interaction, and computing its significance using permutation test. The authors propose two tests based on Lancaster interaction and Sreitberg interaction. The authors demonstrate the effectiveness of these measures on simulated data with higher order interaction and on real data from fMRI measurements.

**Strengths:**

The paper addresses an interesting and relevant problem since estimating higher order interaction provides invaluable insight into the structure and relations among multivariate systems. The paper is very well written; the authors motivates and describes the problem well, and discusses the proposed estimator in detail.

**Weaknesses:**

The paper only discusses the performance of the proposed measure against a kernel based test of independence rather than existing tests on Lancaster and Streitberg interaction.

In the experimental section it is shown the neuroimage dataset shows higher order interaction within a region than between region. It will e great to get some more insight on, e.g., the implication of higher 4-way interaction that 3-way in SOM and higher 3-way interaction than 4-way interaction in DMN. Similarly, the lack on higher order interactions in LLM despite 2-way interactions being prevalent.

**Questions:**

- equation 4, do we need the inequality under the sum given the zeta function?
- equation 5, what is phi_iz

**Limitations:**

The authors address the limitation of the work in section 7, e.g., around computational complexity.

---

> ### Author Rebuttal · Authors · 2023-08-09
>
> We thank the reviewer for their comments and positive response to our paper.
>
> Regarding the suggestion to compare against non-kernel implementations of Lancaster and Streitberg tests, we are not aware of any such alternative approaches. However, we would be open to exploring alternatives for comparing and validating our implemented tests if the reviewer has suggestions.
>
> Although our focus here was on the theory and the neuroimaging data was presented very concisely, we were also intrigued by the analysis of the this data set and its interpretations. If we consider the SOM or VIS, our results show a larger percentage of 2-way interactions compared to regions such as the FPN or DAN, suggesting that there is increased redundancy in the system relative to these other regions. The SOM and VIS are structurally coupled, modular sensorimotor processing regions, which could benefit from information redundancy to increase robustness. These findings are in line with recent work reporting higher redundancy in sensorimotor regions [1], although further investigation is needed to draw firm conclusions about brain function from these analyses.
>
> The potential for future investigation of the high-order features in neural data is large but this fell beyond the limits of the current paper. As an additional investigation, we hierarchically constructed hypergraphs from the identified interactions at increasing order $d$, by defining the high-order interactions as hyperedges and individual regions as nodes, incrementally including the interactions at a higher order. We then analysed some structural properties of these hypergraphs, and report the degree assortativity of each hypergraph (Fig 3, attached). We found that most regions display a negative degree assortativity that became less negative with increasing order interactions.
>
> Additionally, we noticed that the degree assortativities when only considering pairwise connections d=2 are quite similar across regions. Standard deviations of degree assortativities across RSNs are shown in the table below.
> | Degree assortativity | Standard deviation |
> |----------------------|--------------------|
> | 2-way                | 0.044              |
> | 3+2-way              | 0.161              |
> | 4+3+2-way            | 0.104              |
> | 5+4+3+2-way          | 0.059              |
>
> However, when including high-order interactions in the hypergraphs, we find that the degree assortativities of each region diverge from each other. We interpret this as tentative evidence that the structure of macroscopic brain organisation may differ substantially when taking into account interactions beyond pairs, highlighting the importance of methods like ours to reveal new insights from brain data when taking into account high-order interactions that could be of importance for functional processing.
>
> We also thank the reviewer for spotting the type error in Equation 4. You are right that the inequality has been encoded in the zeta function already. We have corrected this in the revised manuscript.
>
> Regarding your question on Equation 5, we are assuming that by $phi\_iz$ you are referring to $\phi^i$? $\phi^i$ is the feature map of variable $X^i$ and $k^i(x, x') = \langle \phi^i (x), \phi^i (x') \rangle$. We apologise for not making this clear in the text and have stated this more clearly in Definition 1 and Definition 2 in the revised manuscript.
>
> [1]  Luppi, A.I., Mediano, P.A.M., Rosas, F.E. et al. A synergistic core for human brain evolution and cognition. Nat Neurosci 25, 771–782 (2022).

---

> > ### Comment · Reviewer_vitW · 2023-08-20
> > **Thank you for your comments**
> >
> > I would like to thank the authors for the detailed comments and discussion on the potential meaning of the higher order interactions.
> >
> > Regarding the standard test, I was wondering if one can take the squared distance between the RHS and LHS of Eqs. (2) and (3) over the support of the distributions, e.g., using Parzen estimate or assuming Gaussian distributions, as a test statistic.
> >
> > Was Fig. 3 attached in the response?

---

> > > ### Author Response · Authors · 2023-08-21
> > > **Reply to second comment**
> > >
> > > We would like to thank the reviewer for the ongoing discussion, and for the additional clarification to their original question regarding standard tests.
> > >
> > > Re: standard tests --- As suggested by the reviewer, comparisons to standard approaches to model distributions and ensuing statistical tests would be worthy of further investigation. However, we can already provide a first answer and discussion based on results from the literature.
> > >
> > > As already noted, the use of kernel-based methods is predicated on their generality (i.e., fewer assumptions due to their being non-parametric), as well as higher test power and reduced 'curse of dimensionality' for high-dimensional data.
> > >
> > > Given the usual lack of knowledge of the underlying distribution in numerous real-world data sets,
> > > imposing assumptions about Gaussianity (or other parametric distributions) can severely limit the applicability of tests and lead to inaccurate conclusions. Indeed, several of our examples are non-Gaussian.
> > >
> > > Going beyond such assumptions, a first strategy to circumvent some of these limitations is by fitting a mixture of Gaussians, often implemented through the Expectation-Maximization (EM) algorithm. However, this poses heavy computational challenges of its own as it is a non-convex problem that might converge slowly to local optima, and with intrinsic problems in establishing the number of Gaussians in the mixture. In particular, Ref [1] showed that the approximation with mixtures of Gaussians is slow and inaccurate for high-dimensional problems.
> > >
> > > Therefore, alternative non-parametric approaches to model arbitrary distributions with better numerical properties have been developed. One of the most widely used is the Parzen estimate suggested by the reviewer, which is also known as kernel density estimation (KDE). However, detailed numerical comparisons have already shown that KDE is not suitable for high-dimensional data and it typically does not consider the dependence structure among the variables [1].
> > > Specifically, Ref [1] showed that a two-sample test based on KDE using the L2 distance has less power compared with the Maximum Mean Discrepancy formulated by kernel mean embedding (as we do in our paper). Indeed, kernel mean embeddings do not suffer from the curse of dimensionality as the rate of convergence of the empirical kernel mean embedding to the true kernel mean embedding of the underlying distribution is independent of the dimensionality. One can (with high probability) obtain an approximation within
> > > $\mathcal{O}(n^{-1/2})$ of the true kernel embedding based on a finite sample of size $n$.
> > >
> > > In summary, kernel-based methods exhibit substantially improved computational and theoretical properties relative to KDE and other non-parametric methods, as well as avoiding any assumptions (e.g., Gaussianity) about the underlying distributions.
> > >
> > > We thank the reviewer for pointing out this comparison and, if accepted, we will include this discussion in our revised manuscript.
> > >
> > > [1] Song, Le, Kenji Fukumizu, and Arthur Gretton. "Kernel embeddings of conditional distributions: A unified kernel framework for nonparametric inference in graphical models." IEEE Signal Processing Magazine 30.4 (2013): 98-111.
> > >
> > > Re: Figure 3 --- Apologies for not being clear: Figure 3 referred to the attached pdf with additional figures that have been produced to support this review process. The PDF is located in the general response at the top. You can find the link right below the tables.

---

### Official Review · Reviewer_KEns · 2023-07-11

**Soundness:** 3 good
**Presentation:** 3 good
**Contribution:** 3 good
**Rating:** 6
**Confidence:** 2

**Summary:**

The authors conduct a very systematic study of tests that measure couplings between variables where these couplings include higher-order interactions. Authors uncover connections with lattice theory that then leverage them to formulate these tests more efficiently and in a more interpretable way. The authors demonstrate empirically that one of these tests, the Lancaster test, is better than dHSIC to test for joint and marginal independence and that the Streitberg test is overall superior to the other tests when it comes to detecting all the factorizations of the joint distribution. Finally, the authors validate their results numerically with synthetic data and brain data.

**Strengths:**

- **originality**: The authors provide novel theoretical connections with lattice theory, which help derive the interaction measures and their statistical tests. These contributions are novel to the best of my knowledge and are noteworthy.

- **quality**: The theoretical derivations seem sound and there are no obvious errors I could identify. The theoretical results are validated with a sufficient amount of simulated and real-world data.

- **clarity**: the paper is very clearly written.



**Weaknesses:**

- **significance**: the significance of the approach is hindered by the fact that it is, as claimed by the authors, "computationally expensive". Additionally, the authors themselves also point out that the theoretical results rely on the assumption of iid data, which limits the applicability of the method to real data.

**Questions:**

No questions for the authors.

**Limitations:**

The authors have addressed adequately all the limitations and negative impact of their work.

---

> ### Author Rebuttal · Authors · 2023-08-09
>
> We thank the reviewer for their review and comments. The theoretical results in our paper leveraging lattice theory enable us to define and implement kernel-based $d$-order Lancaster and Streitberg interaction tests, whose computational complexity was previously prohibitive, thus opening up their practical use for real-world applications.
>
> The original time complexity for computing the Streitberg interaction estimator is $\mathcal{O}(B_d^2 n^{2d})$ where $B_d$ is the Bell number which represent the number of partitions in a set of cardinality $d$ and, consequently,  $B_d^2$ represent the number of terms in the Streitberg interaction estimator. For fixed $n$, our lattice formulation allows us to reduce the number of terms in the mixed cumulant operator from $B_d$ to $F_d$ where $F_d$ is the number of partitions without singletons, equivalent to centring. In Table 3 in the main response, we illustrate the reduction in the number of terms before and after eliminating the partitions with singletons. After this reduction, the complexity becomes $\mathcal{O}(F_d^2 n^{2d})$. For fixed $d$, we can further optimise the time complexity by optimal contraction ordering, using the fact that two index sets are disjoint such that $\mathcal{O}(n^{2d})$ becomes $\mathcal{O}(n^{min(|\pi_s|, |\pi_s'|) + 1})$. Examples of this reduction for fixed $d$ can be found in Table 1 in the main response.
>
> Similarly, the time complexity for computing the $d$-order Lancaster interaction estimator na\"ively is $\mathcal{O}(2^{d+1} n^{2d})$. By centring, i.e. eliminating the partitions with singletons, the only element left is $\hat{1}$. Therefore the Lancaster interaction estimator can be computed in $\mathcal{O}(dn^2)$.
>
> Additional examples with stationary time series data:
> Following the reviewer's comment on $iid$ data, we show that our $d$-order interaction tests can be generalised to time-series data too. The tests introduced here can be used in conjunction with a permutation procedure that approximates the null distribution by shifting through the time observations following a recent paper [1]. We are thus able to test high-order interactions between $d$ stationary random processes.
> We have implemented this strategy and show some results from synthetic data in Figure 4 in the attached pdf. Again, we see that the Lancaster test is able to detect factorisations that corresponds to the partitions with at least one singleton, and the Streitberg test is able to detect any factorisations (dHSIC failed in both a and b). This means that both tests have controlled type I error. In Figure 4c, we also see that the Streitberg and Lancaster tests reach full power with fewer time observations compared with dHSIC when testing joint independence. Similarly when testing high-order interactions in Figure 4d, Streitberg is more data-efficient compared with modified dHSIC.
>
> [1] Liu, Zhaolu, et al. "Kernel-based Joint Independence Tests for Multivariate Stationary and Nonstationary Time-Series." arXiv preprint arXiv:2305.08529 (2023).

---

### Official Review · Reviewer_SriA · 2023-07-13

**Soundness:** 3 good
**Presentation:** 2 fair
**Contribution:** 2 fair
**Rating:** 5
**Confidence:** 2

**Summary:**

The paper is a nice introduction to measuring high order interactions between groups of random variables and an experimental demonstration of measuring these interactions for up to 4 variables inclusively. The main focus is on the Lancaster and Streitberg interactions tests via kernel methods such as the Hilbert Schmidt Information Criterion. After presenting a mix of prior and, possibly, novel results some experimental evidence is provided that the approach works.


**Strengths:**

1. The topic of capturing interactions beyond binary is interesting and largely avoided for lack of adequate tools.
2. The paper is generally well written and the didactic value is great. It can serve as an accessible introduction into the topic.


**Weaknesses:**

1. The paper's contribution is unclear. Some of the claims from the abstract are already known previously published work, such as the link to the lattice theory (Streitberg) and the work with d=4 [33, 34].
2. The paper claims efficiency of the described permutation procedure but besides high level considerations of its asymptotic behavior no experimental characterization is provided. Especially the interplay of efficiency and accuracy. When the procedure is supposed to be faster for a smaller number of samples how fast does the accuracy drop?
3. I found the presentation unclear. On one hand extreme novelty and generality with respect to d is claimed. On the other, experiments are only shown for d up to 4 (previously available results).


**Questions:**

1. What is the single technical novel result in this paper?
2. Experiments demonstrating the effect of the interplay between the accuracy and computational complexity. Computational complexity numbers reported for all runs of the fMRI experiments.
3. Error bars in Figure 3 are missing. Are results so stable?
(the authors have responded to my questions, which is now reflected in my raised score)

**Limitations:**

nothing to report

---

> ### Author Rebuttal · Authors · 2023-08-09
>
> We thank the reviewer for their comments.
>
> - **Novelty**: Although, as the reviewer points out, the mathematical link between lattice theory and the $d$-order probabilistic Streitberg measure was indicated in [25] and briefly mentioned in [33], in neither reference were interaction tests implemented owing to their computational complexity. (Reference [34] does not make any references to lattice theory.) Regarding novelty, the main theoretical contribution here is to leverage lattice theory to formulate and optimise $d$-order Lancaster and Streitberg interaction tests that are computable on real data, thus enabling practical applications. More specifically:
> First, we formulate the differences between joint independence, Lancaster interaction and Streitberg interactions in terms of the partition lattice and its sublattices.
> Second, we define the $d$-order Streitberg and $d$-order Lancaster interaction tests (with improved understanding of the vanishing conditions on the Lancaster interaction) in the kernel setting, and optimise their computational efficiency by eliminating partitions with singletons in the respective lattices.
> Third, we propose a recursive computation strategy such that one would only need to compute the inner product if the join of the two partitions is $\hat{1}$.
> Fourth, after the optimisation using centring, the Hilbert-Schmidt norm of each high-order measure can be directly obtained from the product lattice (explained in detail in section B of the SI).
> Finally, we show that using the interval lattice we can derive a generalised interaction measure whose test formulation can be directly obtained from the 2nd level of the lattice which captures the corresponding factorisations in the sub-hypotheses.
>
> - **Computational efficiency**: We apologise for our lack of clarity. Our comment about efficiency relates to the fact that instead of having to consider all possible factorisations ($B_d$, Bell number of $d$ variables), we only need to perform $(2^{d-1}-1)$ sub-tests of factorisations corresponding to the 2nd level of the partition lattice, since all other sub-tests are consequences of these. This is a substantial reduction in computation that follows from the lattice formulation. We have corrected our wording in the main text to make this clearer.
>
> - **Interplay of efficiency and accuracy**: We thank the reviewer for this remark. To explore this, we have performed 4 different experiments each with 3 different interaction strengths to show how decreasing the sample size affects the null rejection rate (Fig 1, attached). In all cases, as the sample size decreases, the null rejection rates of dHSIC decay much faster than both Lancaster and Streitberg.
>
> - **Higher order $d>4$ and practical applications**: To highlight the generality of the proposed statistical tests to any order, we have performed $d=5$ numerical experiments for both the synthetic examples (Fig 2, attached) and the real-world data (Fig 3, attached). The decay of null rejection rate with decreasing sample size for $d=5$ is included in Fig. 1.
>
> - **Our empirical results provide backing for  practical applications**: We have shown that: (i) Lancaster is better than dHSIC for testing joint and marginal independence; and (ii) Streitberg is more data-efficient compared to dHSIC for detecting high-order interactions. Moreover, we show that these tests are practical to compute on standard computers for iid data, and we have now added further examples of applications to stationary time-series data (Fig 4, attached).
>
> - **Computational complexity**: The computational complexity of dHSIC, Lancaster and Streitberg tests for $d=2,3,4,5$ are shown in Table 1 in the main response. For $d=4,5$, we show both the full complexity of the na\"ive enumeration compared to the substantially reduced complexity after optimisation using the lattice formulation. We also show the computational time for all fMRI experiments in Table 2. All experiments carried out on a 2015 iMac with 4 GHz Quad-Core Intel Core i7 processor and 32 GB 1867 MHz DDR3 memory.
>
> - **Error bars**: Thank you for spotting this error. Error bars in the bar chart now added to Figure 3. The results are stable, and this will be changed in the revision.

---

> > ### Comment · Reviewer_SriA · 2023-08-13
> > **computational complexity**
> >
> > Thank you for your clarifications and the additional tables. It would be unfortunate, if the tables and the additional plot do not make it either to the main manuscript of the supplement of the paper. I hope this additional work is included in the manuscript if it gets accepted.
> >
> > My doubt on the real-world applicability still holds. Don't you need to compute the proposed measure on all d-element groups of your 100 variables in order to find the ones that do d-interact?
> >
> > What about the case, when we have considered all triplets for d=3 and found those that do interact in 3-way. Do we eliminate them from consideration when considering d=4? What if the variables that 3-way interact are a part of a larger clique?
> >
> > I am raising my score to acknowledge your previous clarifications.

---

> > > ### Author Response · Authors · 2023-08-15
> > > **Re: computational complexity**
> > >
> > > We thank the reviewer for their comments, questions and score change. If accepted, we will certainly include in the revised manuscript all the additional work (figures, tables, explanations, discussion) in these responses.
> > >
> > > As the reviewer points out, the problem of detecting $d$-order interactions among a group of $N$ variables is combinatorial, as it entails checking the groups of $d$ variables chosen from the $N$ variables. Clearly, such combinatorial problems become infeasible as $N$ and/or $d$ become large. However, recent work on real data from various application areas has focused on revealing any interactions beyond pairwise, i.e., $d>2$. It has been shown that low order interactions ($d=3,4$) already make a significant difference to network structure and network dynamics [1][2][3],  highlighting the potential benefits to be gained from higher order interaction tests.
> > >
> > > Furthermore, many well-recognised and widely used theoretical approaches only have closed forms for $d=3$ [4][5], and cannot be generalised to arbitrary $d$ since the number of terms in those approaches is related to the Dedekind number which becomes rapidly intractable [6]. In contrast, our paper shows that the Streitberg interaction can be explicitly defined for any $d$, and we devise theoretical and computational strategies to reduce its computational cost via the lattice theory formulation.
> > >
> > > We thank the reviewer for the second comment on the recursive aspects of the computation of $d$-order interactions. Indeed, it is naturally possible to discount the computation of certain higher-order interactions when some lower-order interactions are present. If all the lower order interactions are absent, then testing the $d$-order interaction is the same as testing the joint independence for $d$ variables. For example when $d=3$ and all pairwise interactions are absent, then testing the Streitberg interaction reduces to testing joint independence of 3 variables. More generally, if we test the interactions bottom-up, i.e., recursively from the lower orders upwards, the expression for Streitberg becomes simpler whenever there is a lower-order independence. The simplification is possible because the Streitberg interaction can be rewritten as the sum of the differences between a factorisation and the product of the marginals due to the fact that the sum of Mobius coefficients is zero. Hence the presence of a lower order independence allows us to simplify the $d$-order interaction formula.
> > >
> > > Alternatively, there are also cost-reducing simplifications if we do the tests top-down (i.e. starting from order $d$ downwards), although the problem still remains combinatorial. If there are partial rejections for some tests involved in the $d$-order Streitberg test, we can narrow down the possible choices of the factorisation (as discussed in Section C of SI). This allows us to eliminate the lower order factorisations that fall in the lattice branches of the rejected 2nd level factorisations, thus reducing the total tests needed for the factorisation of the $d$-variables. Further simplifications are achieved by accounting for overlaps of the lattice branches.
> > >
> > > Your comment on lower-order interactions that may be part of a larger clique also lands on an important direction that complements the above points on recursive efficiency. Let us consider a system whose interactions have been identified using the Streitberg interaction tests up to order $d$, as we have shown for the fMRI data set. This allows us to build a hypergraph (see paper and the additional computations and figures in attached pdf).  One could then interpret the hypergraph as follows: i) If a $d$-order hyperedge is detected and none of the lower order $(d-1)$-order hyperedges are present then we say that this $d$-order hyperedge reflects a purely synergistic (or emergent) interaction between the $d$ variables; ii) if the $d$-order and all the lower order interactions are present, then we say that the $d$-order interaction is purely redundant (and corresponds to a simplicial complex construction); iii) if some, but not all, lower order interactions are present (e.g., for 4 variables, only the 4-way interaction and one 3-way interaction are present), then both synergy and redundancy are present in this group of variables. Such a construction could offer an alternative, statistically-motivated approach to computing synergy and redundancy, an important current topic of research in computational neuroscience.
> > >
> > > - [1] Battiston et al. Physics Reports 874 (2020): 1-92.
> > > - [2] Santoro et al. Nature Physics 19.2 (2023): 221-229.
> > > - [3] Varley et al. PNAS 120.30 (2023): e2300888120.
> > > - [4] Ince, arXiv:1702.01591 (2017).
> > > - [5] Williams and Randall, arXiv:1004.2515 (2010).
> > > - [6] Van Hirtum et al. arXiv:2304.03039 (2023).

---

> > > > ### Comment · Reviewer_SriA · 2023-08-16
> > > > **combinatorial optimization**
> > > >
> > > > Thank you for an interesting discussion. I appreciate your effort in putting this specific commentary together - thought provoking.

---

### Author Rebuttal · Authors · 2023-08-09

We thank the reviewers for their clear reviews and thoughtful questions. In addition to the specific answers in the responses to each reviewer, we would like to briefly address here three overarching themes that have appeared in the reviews: the computational complexity of our method; implications of the analysis of real-world neuroimaging data; and the novel contributions of this manuscript above and beyond prior work. We address each of these in turn:

- **Computational complexity**. As previously stated, our method scales combinatorially in $d$, as would be expected for a method that evaluates $d$-order interactions. Nonetheless, for finite $d$ values relevant in practical applications our method achieves substantial and measurable improvements in complexity (see Table 1 below).
This was achieved by three technical contributions:
    1. elimination of partitions with singletons via kernel centring;
    2. formulation of an optimal contraction ordering; and
    3. a recursive computational strategy.

    These reductions follow from our mathematical formulation and links to lattice theory. Moreover, we run various experiments and highlight the practicality of implementing these tests with different sample sizes and also with different data types (i.e., stationary time series) which were previously unattainable without our theoretical improvements. Some of this work was included in the SI, and additional work is explained in the individual responses below. In the revised version of the paper, we will include further discussion of prior work and add experimental results to show explicitly that the ways to optimise using lattice theory are novel leading to substantial reductions in computation.
- **Implications of the results in real-world data**. The key take-away message from our experiments --- beyond the fact that the proposed quantities are tractable --- is that brain activity has substantial and heterogeneous high-order structure that can be revealed using our method. This high-order structure is non-trivial, in the sense that it cannot be predicted from pairwise interactions, and calls for new hypergraph analyses to understand brain function across scales. In the revised version of the paper we will report new experiments emphasising the importance of this high-order structure and discuss links with supporting neuroscientific evidence.

- **Novelty**. The paper's main novel theoretical contribution is the use of lattice theory to formulate and optimise $d$-order Lancaster and Streitberg interaction tests. The novel contributions include:
    1. We use the partition lattice and its sublattices to make a qualitative comparison between the information encoded in joint independence, Lancaster interaction and Streitberg interactions.
    2. We formalise the $d$-order Streitberg and the $d$-order Lancaster in the kernel test setting and optimise their computational efficiency by eliminating the partitions with singletons in the respective lattices.
    3. We propose a recursive computation strategy such that one would only need to compute the inner product if the join of the two partitions is $\hat{1}$.
    4. After the optimisation using centring, the Hilbert-Schmidt norm of each high-order measure can be directly obtained from the product lattice (explained in detail in section B of the SI).
    5. We show that using the interval lattice we can derive a generalised interaction measure. The respective test formulation is also related to lattice since the corresponding factorisations in the sub-hypotheses can be directly obtained from the 2nd level of the lattice.

**Table 1**. Time complexity for $d$-order interaction estimators.
| $d$-order | dHSIC              | Lancaster$\Rightarrow$(optimised)                      | Streitberg$\Rightarrow$(optimised)                     |
|-----------|--------------------|--------------------------------------------------------|--------------------------------------------------------|
| 2-way     | $\mathcal{O}(n^2)$ | $\mathcal{O}(n^2)$                                     | $\mathcal{O}(n^2)$                                     |
| 3-way     | $\mathcal{O}(n^2)$ | $\mathcal{O}(n^2)$                                     | $\mathcal{O}(n^2)$                                     |
| 4-way     | $\mathcal{O}(n^2)$ | $\mathcal{O}(n^8)$$\Rightarrow${$\mathcal{O}(n^2)$}    | $\mathcal{O}(n^8)$$\Rightarrow${$\mathcal{O}(n^3)$}    |
| 5-way     | $\mathcal{O}(n^2)$ | $\mathcal{O}(n^{10})$$\Rightarrow${$\mathcal{O}(n^2)$} | $\mathcal{O}(n^{10})$$\Rightarrow${$\mathcal{O}(n^3)$} |

**Table 2**. Time for experiments in Fig 3.
|     | SOM      | VIS     | SAL   | DAN    | DMN     | FPN    | LIM | Random |
|-----------|----------|---------|-------|--------|---------|--------|-----|--------|
| 2-way     | 1s       | 2s      | 1s    | 2s     | 2s      | 2s     | 1s  | 12s    |
| 3-way     | 12s      | 18s     | 7     | 13s    | 8s      | 8s     | 1s  | 13s    |
| 4-way     | 5m36s    | 5m      | 2m44s | 2m31s  | 2m      | 2m10s  | 1s  | 2m41s  |
| 5-way     | 2h18m24s | 1h30m9s | 1h3m  | 54m40s | 1h12m4s | 27m16s | 2s  | 29m    |

**Table 3**. Number of terms in the test statistics before and after eliminating the partitions with singletons.
| # of var                 | 4                  | 5                   | 6                    | 7                     | 8                      | 9                        |
|----------------------------------|--------------------|---------------------|----------------------|-----------------------|------------------------|--------------------------|
| Streitberg$\Rightarrow$(centred) | 15$\Rightarrow$(4) | 52$\Rightarrow$(11) | 203$\Rightarrow$(41) | 877$\Rightarrow$(162) | 4140$\Rightarrow$(715) | 21147$\Rightarrow$(3425) |
| Lancaster$\Rightarrow$(centred)  | 12$\Rightarrow$(1) | 27$\Rightarrow$(1)  | 58$\Rightarrow$(1)   | 121$\Rightarrow$(1)   | 248$\Rightarrow$(1)    | 503$\Rightarrow$(1)      |

---

### Decision · Program_Chairs · 2023-09-21

**Decision:**

Accept (poster)

**Comment:**

This paper considers the problem of how to take higher-order interactions into account in statistical modeling, and proposes a hierarchy of d-order interaction measures with kernel-based nonparametric tests to assess their statistical significance. It is also remarkable to draw a connection to the lattice theory.

This paper is overall well-written, and all the reviewers are happy with the contribution of the paper and recommend acceptance. I also agree with it.

I have one comment about related work. This paper has a close relationship to information geometry of hierarchies of probability distributions, which is not discussed in the paper. Information geometry of statistical models with higher-order interactions has been established in a seminal paper [1],  where it is shown that natural parameters of the log-linear model with higher-order interactions and its expectations form mutually orthogonal coordinate systems of (dually flat) statistical manifolds.
Moreover, recently, this line of research has been further generalized into any posets using Zeta/Möbius transformation in Section 4 in [2], where the lattice theory has been already used in probabilistic analysis.
Since they are quite relevant to the contribution of this paper, I strongly recommend adding some discussion in the camera-ready version of the paper.

To summarize, I would appreciate the contribution of the paper, which is sound and novel. Therefore, I recommend acceptance of the paper.

[1] S. Amari. Information geometry on hierarchy of probability distributions. IEEE Transactions on
Information Theory, 47(5):1701–1711, 2001.
[2] M. Sugiyama,  et al.: Tensor Balancing on Statistical Manifold, ICML2017.